# Adaptative Divergence of *Cryptococcus neoformans*: Phenetic and Metabolomic Profiles Reveal Distinct Pathways of Virulence and Resistance in Clinical vs. Environmental Isolates

**DOI:** 10.3390/jof11030215

**Published:** 2025-03-12

**Authors:** Camila Botelho Miguel, Geovana Pina Vilela, Lara Mamede Almeida, Mariane Andrade Moreira, Glicélia Pereira Silva, Jamil Miguel-Neto, Melissa Carvalho Martins-de-Abreu, Ferdinando Agostinho, Javier Emilio Lazo-Chica, Mariana Santos Cardoso, Siomar de Castro Soares, Aristóteles Góes-Neto, Wellington Francisco Rodrigues

**Affiliations:** 1Multidisciplinary Laboratory of Scientific Evidence, University Center of Mineiros (Unifimes), Mineiros 75833-130, GO, Brazil; camilabotelho@unifimes.edu.br (C.B.M.); gepinavilela@academico.unifimes.edu.br (G.P.V.); laramamede@academico.unifimes.edu.br (L.M.A.); marianemoreira1102@academico.unifimes.edu.br (M.A.M.); glicelia@unifimes.edu.br (G.P.S.); jamil@unifimes.edu.br (J.M.-N.); dramelissa@unifimes.edu.br (M.C.M.-d.-A.); 2Postgraduate Program in Tropical Medicine and Infectious Diseases, Federal University of the Triângulo Mineiro—UFTM, Uberaba 38025-180, MG, Brazil; siomar.soares@uftm.edu.br; 3Faculty of Physiotherapy, University of Rio Verde, UniRv, Rio Verde 75901-970, GO, Brazil; ferdinando@unirv.edu.br; 4Laboratory of Cell Biology, Department of Structural Biology, Institute of Biological and Natural Sciences, Federal University of Triângulo Mineiro (UFTM), Uberaba 38061-500, MG, Brazil; javier.chica@uftm.edu.br; 5Molecular and Computational Biology of Fungi Laboratory, Department of Microbiology, Instituto de Ciências Biológicas, Universidade Federal de Minas Gerais, Belo Horizonte 31270-901, MG, Brazil; marianascardoso@yahoo.com.br (M.S.C.); arigoesneto@gmail.com (A.G.-N.)

**Keywords:** *Cryptococcus neoformans*, phenetic analysis, metabolomics, clinical isolates, environmental isolates, antifungal resistance, virulence adaptation

## Abstract

*Cryptococcus neoformans* is a life-threatening fungal pathogen that primarily affects immunocompromised individuals. While antiretroviral therapy has reduced incidence in developed nations, fluconazole-resistant strains and virulent environmental isolates continue to pose challenges, especially because they have many mechanisms of adaptability, supporting their survival. This study explores the phenetic and metabolomic adaptations of *C. neoformans* in clinical and environmental contexts to understand the factors influencing pathogenicity and resistance. Methods: An in silico observational study was conducted with 16 *C. neoformans* isolates (6 clinical, 9 environmental, and 1 reference) from the NCBI database. Molecular phenetic analysis used MEGA version 11.0.13 and focused on efflux pump protein sequences. Molecular phenetic relationships were assessed via the UPGMA clustering method with 1000 bootstrap replicates. The enzymatic profiling of glycolytic pathways was conducted with dbCAN, and metabolomic pathway enrichment analysis was performed in MetaboAnalyst 6.0 using the KEGG pathway database. Results: Molecular phenetic analysis revealed distinct clustering patterns among isolates, reflecting adaptations associated with clinical and environmental niches. Clinical isolates demonstrated enriched sulfur metabolism and glutathione pathways, likely adaptations to oxidative stress in host environments, while environmental isolates favored methane and glyoxylate pathways, suggesting adaptations for survival in carbon-rich environments. Conclusion: Significant phenetic and metabolomic distinctions between isolates reveal adaptive strategies for enhancing virulence and antifungal resistance, highlighting potential therapeutic targets.

## 1. Introduction

Cryptococcosis is a life-threatening fungal infection caused primarily by *Cryptococcus neoformans*, an opportunistic pathogen responsible for severe diseases such as meningitis, particularly in immunocompromised individuals. This infection remains a significant global health burden, with approximately 223,100 cases reported annually among patients with HIV, resulting in an estimated 181,100 deaths, predominantly in sub-Saharan Africa [1,2]. While antiretroviral therapy has reduced the incidence of cryptococcosis in developed nations, the emergence of fluconazole-resistant strains and virulent environmental isolates continues to pose challenges for clinical management [3].

The pathogenicity of *C. neoformans* is closely linked to its genetic and phenotypic plasticity, which enables adaptation to diverse ecological niches. Among its key virulence factors is the production of a polysaccharide capsule, composed primarily of glucuronoxylomannan (GXM) and galactoxylomannan (GalXM), which facilitates immune evasion and systemic dissemination [4,5]. GXM inhibits phagocytosis and modulates cytokine production to favor an anti-inflammatory response, while GalXM induces macrophage apoptosis, further weakening the host’s immune defenses [6]. Additionally, melanin synthesis enhances oxidative stress resistance, and thermotolerance allows the pathogen to survive within mammalian hosts [7,8]. The capsule’s multifaceted role in immune evasion underscores its importance as a central virulence factor and a potential therapeutic target.

Beyond its clinical adaptations, *C. neoformans* thrives in environmental reservoirs such as soil, bird guano, and decaying organic material. Its environmental resilience reflects an evolutionary connection between ecological adaptation and clinical virulence [9]. Notably, exposure to agricultural triazole fungicides has been implicated in the development of cross-resistance to clinical azoles, further complicating treatment strategies [10]. Phenetic analyses, including studies on efflux pump proteins, highlight how selective pressures across niches shape the pathogen’s adaptive strategies [9]. Concurrently, metabolomic studies have shown that clinical isolates exhibit enhanced sulfur metabolism and glutathione pathways to counteract oxidative stress in host environments, while environmental isolates prioritize carbon metabolism to thrive in nutrient-variable conditions [3,11].

This study aims to investigate the phenetic and metabolomic adaptations of *C. neoformans* strains isolated from clinical and environmental sources. By integrating molecular phenetic analyses and comprehensive metabolomic profiling, we seek to elucidate the distinct adaptive strategies employed by *C. neoformans* in varied ecological contexts. These findings provide critical insights into the pathogen’s evolution and may inform the development of novel therapeutic strategies to combat cryptococcosis, particularly in the face of rising antifungal resistance [1,9].

## 2. Materials and Methods

### 2.1. Study Description and Ethical Aspects

This is an observational in silico study. Open access databases were consulted, and no national or international legislation on scientific research was violated. No human beings or animals were used directly or indirectly, only information of microbial agents and fungi.

### 2.2. Design and Selection of Strains

Two groups were included in this study: clinical and environmental strains of *C. neoformans*, along with a reference strain (*C. neoformans* var. *grubii* JEC21 was chosen as the reference strain based on its extensive genomic characterization and widespread recognition in the literature as a standard for *C. neoformans* var. *grubii* [12,13]). Six clinical strains (c45, CHC193, MW-RSA1955, Br795, c8, and Th845) and nine environmental strains (VNII, MW-RSA852, A1-35-8, AD1-83a, A5-35-17, D17-1, Ze90-1, Tu259-1, and Gb118) were selected. Including the reference strain JEC21, a total of 16 genomes were analyzed in this study. The reference strain was used as a benchmark but was not included in comparative analyses between the clinical and environmental groups. Variables such as the year of isolation, location (country), and isolation source were recorded to better characterize the collected strains.

We utilized genomes of *C. neoformans* isolates available in the GenBank/National Center for Biotechnology Information (NCBI) database to ensure consistency in data acquisition. The dataset included strains stratified as clinical or environmental, consistent with the study’s objectives. While other strain types, such as CBS 8710, CBS 7229, and CBS 8273 (*C. gattii*), are scientifically relevant, their genomes were unavailable in the selected database, and their inclusion could diverge from the study’s focus on *C. neoformans* adaptations.

Metadata associated with the dbCAN database output, which provides annotations of carbohydrate-active enzymes (CAZymes), as well as descriptions of relative abundance values across strains and virulence data, have been previously deposited in the OSF Home repository, which serves as a publicly accessible platform for storing and sharing metadata and research data, ensuring the transparency and reproducibility of the analyses performed in this study. The complete dataset is available for review at the following link: https://doi.org/10.17605/OSF.IO/8JF2K (accessed on 15 January 2025).

### 2.3. Molecular Phenetic Analysis

The molecular phenetic analysis was conducted using MEGA software (v11.0.13) to ensure the precise reproducibility of analytical steps. For this analysis, we specifically selected the multidrug efflux pump protein from environmental and clinical strains of *C. neoformans* due to its well-documented role in antifungal resistance and environmental adaptation.

Our approach was hypothesis-driven, aiming to determine whether this resistance-associated protein exhibited structured differentiation between clinical and environmental isolates, rather than attempting a whole-genome phylogenetic reconstruction. The rationale for this choice is that whole-genome phylogenetic approaches could introduce confounding variations unrelated to drug resistance, making it more challenging to isolate adaptive signals specific to antifungal resistance mechanisms. Additionally, multi-locus sequence typing (MLST), while valuable for strain classification, does not directly capture protein-level adaptive responses.

To ensure methodological rigor, protein sequence alignment was performed using ClustalW v2.1, optimizing alignment quality for subsequent analyses. Evolutionary distances among sequences were then estimated using the Dayhoff substitution model, and the robustness of these estimates was assessed with the bootstrap method (1000 replications) to enhance result reliability.

The resulting distance matrix was used to reconstruct the molecular phenetic tree, employing the UPGMA clustering method. The tree was further validated through a bootstrap test (1000 replicates), producing a consensus tree that represents the evolutionary relationships among environmental and clinical strains of *C. neoformans*. These analyses allowed us to assess whether resistance-related phenetic divergence occurs between clinical and environmental isolates, supporting our initial hypothesis.

### 2.4. Carbohydrate Pattern Analysis of Enzymes

To investigate potential differences in carbohydrate patterns between environmental and clinical subgroups of *C. neoformans*, an analysis was conducted using the dbCAN3 program with the HMMER profile (E-value < 1 × 10^−15^, coverage > 0.35), utilizing the latest HMMdb v13 (14 August 2024). This analysis enabled the identification and characterization of enzymes involved in carbohydrate metabolism for each subgroup. Enzyme frequencies were assessed within each genomic subpopulation of environmental and clinical *C. neoformans* isolates.

After enzymes were matched across the two subgroups, the frequency relationships were calculated, with differences equal to or greater than 10% serving as criteria for metabolic predictions. Frequencies showing higher abundance in clinical samples were marked for the clinical subgroup, and the reverse was applied to the environmental subgroup.

### 2.5. Metabolomic Evaluations

After glycolytic enzyme patterns with higher abundance were identified in both clinical and environmental isolates, the corresponding query ID codes were retrieved from the National Center for Biotechnology Information (NCBI) database, and their amino acid residue sequences were extracted in FASTA format. Two datasets were created and subsequently uploaded to the Human Metabolome Database (HMDB) (https://hmdb.ca/) (accessed on 23 August 2024). Metabolite interactions were analyzed separately for both categories: those with higher abundance in the clinical subgroup and those in the environmental subgroup. The metabolic pathway analysis focused on metabolites exhibiting distinct abundance patterns in each subgroup, allowing for a targeted investigation of potential adaptive metabolic responses.

To conduct metabolic pathway enrichment analysis, we used MetaboAnalyst 6.0 (https://www.metaboanalyst.ca/) (accessed on 23 August 2024). HMDB codes corresponding to selected enzymes from *C. neoformans* isolates were classified based on their association with either clinical or environmental samples. The Hypergeometric test was employed for the enrichment analysis, using Relative Betweenness Centrality as the topological measure. To ensure comprehensive pathway representation, the reference library setting “Use all compounds in the selected pathway library” was applied. The KEGG pathway database was used as the primary reference, integrating data specific to *C. neoformans* available as of December 2023. Results were visualized as scatter plots, highlighting significantly enriched pathways and their respective centralities.

A generalized linear model was applied to calculate the Q-stat for each metabolite set, where the Q-stat represents the average of Q values computed for individual metabolites within each pathway. In this context, the Q value quantifies the squared covariance between each metabolite and the observed metabolic adaptation, enabling a refined assessment of pathway-level metabolic differences.

Although this approach provides valuable insights into the metabolic landscape of *C. neoformans*, it is important to note that our analysis was limited to genomic sequence data. Consequently, we did not assess gene copy number variations (CNVs), post-transcriptional regulatory mechanisms (e.g., alternative splicing, mRNA stability, protein degradation), or single-nucleotide polymorphisms (SNPs) between clinical and environmental strains. While these factors play a crucial role in modulating enzyme activity and metabolic adaptation, their evaluation would require transcriptomic (RNA-seq) or proteomic datasets, which are not consistently available for all isolates in public repositories.

Despite these limitations, our metabolomic enrichment analysis offers a functional prediction of metabolic adaptations, suggesting potential regulatory mechanisms influencing metabolic divergence between clinical and environmental isolates. Future research integrating multiomics approaches (genomics, transcriptomics, and proteomics) will be essential to further validate and expand upon our findings.

### 2.6. Virulence Profile Analysis of C. neoformans Isolates

Virulence factors in *C. neoformans* isolates were identified using the PHI-base database (http://www.phi-base.org/, accessed on 25 October 2024), a resource cataloging virulence genes across a broad range of pathogens. This database enabled the annotation of potential virulence factors in the genomic sequences of clinical and environmental isolates. Due to the diverse nature of the database, which includes pathogens with differing infection strategies, the identified factors are considered predictive and not definitive for *C. neoformans*. Further experimental validation would be required to confirm their relevance in this species.

Data on the occurrence of virulence factors were extracted from PHI-base and organized into a frequency matrix for comparative analysis between clinical and environmental isolates. To ensure data consistency, missing values were substituted with zeroes. The comparative analysis aimed to identify frequency differences in potential virulence factors that could reflect adaptive strategies in distinct ecological niches.

The visualization of virulence profiles was performed using heatmaps generated in R (v4.3.2) with the pheatmap package. The heatmap highlighted the distribution and clustering of virulence factors, organized by “ID_group” (clinical or environmental) and “Virulence Profile”. A gradient color scheme (white to green) was applied to represent frequency values, enhancing the interpretability of the data. To complement this, a stacked bar chart was generated using the ggplot2 package in R, illustrating the proportional distribution of virulence profiles across isolate groups.

Statistical associations between virulence profiles and isolate origins (clinical vs. environmental) were assessed using a chi-square test. The statistical analysis yielded a chi-square value of 14.40 (*p* = 0.006), indicating a significant association. These values were incorporated into the bar chart for reference, ensuring clarity and transparency in the presentation of results.

This methodology provided a structured approach to evaluating virulence factors in *C. neoformans*, offering a foundation for exploring ecological adaptations and their potential impact on pathogenicity and antifungal resistance. While the virulence profiles identified here are valuable for hypothesis generation, they remain predictive and necessitate further validation under laboratory conditions.

### 2.7. Data Analysis

The statistical analyses in this study were designed to ensure robustness and reliability, given the complexity and volume of comparisons performed. For the molecular phenetic analysis, evolutionary relationships among the strains were inferred using the UPGMA clustering method in MEGA software (v11.0.13) with the Dayhoff substitution model. To validate these inferences, bootstrap tests with 1000 replicates were applied, and a bootstrap consensus tree was generated to represent the relationships among clinical and environmental isolates [14].

Differences in glycolytic enzyme patterns between clinical and environmental isolates were assessed using dbCAN (HMMER profile; E-value < 1 × 10^−15^, coverage > 0.35). Enzymes exhibiting a relative abundance difference of at least 10% between the groups were selected for further analysis [15]. Enrichment analyses for metabolic pathways were performed in MetaboAnalyst 6.0, applying the Hypergeometric test with Relative Betweenness Centrality as the topological measure. The KEGG pathway database was used as the reference, and results were visualized as scatter plots showing pathway significance and impact [16]. To control for multiple comparisons, the Benjamini–Hochberg method was used to adjust *p*-values, and both raw and FDR-adjusted *p*-values (q-values) were reported [17].

Comparative analyses of virulence profiles were conducted to evaluate associations between ecological niches and the presence of specific virulence factors. Statistical significance was assessed using chi-square tests to determine associations between virulence profiles and isolate origins (clinical vs. environmental) [18]. Visualizations, including heatmaps and bar charts, were created in R (v4.3.2) using the pheatmap and ggplot2 packages, respectively [19,20]. These figures were designed to present statistical insights clearly, with annotations indicating *p*-values, FDR-adjusted q-values, and effect sizes.

All statistical analyses adhered to rigorous bioinformatics and statistical standards to maintain methodological consistency [20]. The significance threshold for all tests was set at *p* < 0.05. Adjusted results were cross-verified with the raw data to ensure consistency and reliability.

## 3. Results

### 3.1. Overview of Selected C. neoformans Strains for Phenetic and Enzymatic Analysis

Table 1 provides details on the *C. neoformans* strains selected as the foundational step for investigating potential adaptation associations between clinical and environmental niches. This dataset includes both clinical and environmental strains, chosen to assess differences in glycolytic enzyme patterns and their involvement in virulence factors and antifungal resistance. The table summarizes key information such as the strain type, specific taxonomic variety, collection date, geographic origin, isolation source, and host. This initial selection of strains was critical for setting up a comparative framework to explore how these organisms may adapt to diverse ecological niches, which could influence their enzymatic profiles and potential for pathogenicity.

### 3.2. Significant Phenetic Differentiation Between Clinical and Environmental Isolates of C. neoformans var. grubii and Evidence of Cross-Adaptation

A molecular phenetic analysis was conducted based on the sequences of the multidrug efflux pump protein of *C. neoformans* var. *grubii* isolates obtained from clinical and environmental samples (Figure 1). The molecular phenetic tree, constructed using the UPGMA clustering method with bootstrap support of 1000 replications, revealed clear differentiation between clinical and environmental samples, suggesting a possible phenetic divergence between these two groups.

The clinical isolates clustered together in a cohesive clade with high bootstrap support (100), indicating high genetic similarity among these lineages. Notably, environmental isolates TU259-1, A1-35-8, and AD1-83 formed a subgroup with robust statistical support (bootstrap of 61), suggesting a close evolutionary relationship among these sequences. Other environmental isolates, such as GB118 and ZE90-1, also exhibited strong molecular phenetic relationships with each other.

In contrast, the clinical isolates were divided into two main subgroups. The first subgroup, composed of MW-RSA1955, TH84, and CHC193, showed strong bootstrap support (100), indicating high cohesion among these samples. The second subgroup, containing isolates C8 and BR795, also demonstrated high bootstrap support (100). Notably, the clinical isolate C45 clustered with the environmental samples, suggesting a possible environmental origin for this isolate or cross-adaptation. However, other potential explanations, such as sampling errors, misidentification, or contamination, cannot be excluded and warrant further investigation (Figure 1).

The reference strain JEC21 was included in the analysis to provide a baseline comparison and was positioned outside the clinical and environmental clades. Additionally, it is important to note that the sequence for the multidrug efflux pump protein was not found for the VNII isolate, and therefore, it was excluded from this molecular phenetic reconstruction.

These findings reinforce the hypothesis that although clinical and environmental samples of *C. neoformans* var. *grubii* share common genetic characteristics, there is a phenetic divergence that may reflect adaptations to the different selective pressures of the isolation environments. The presence of a clinical isolate within the environmental clade also suggests that transmission events from the environment to the clinical context may have occurred.

### 3.3. Metabolic Differentiation and Glycolytic Enzyme Patterns in Clinical and Environmental Isolates of C. neoformans

The significant phenetic differentiation observed between clinical and environmental isolates of *C. neoformans* var. *grubii* (Figure 1), along with indications of cross-adaptation, prompted further investigation into glycolytic enzyme patterns to explore potential associations with resistance and virulence factors. As such, Table 2 presents the results of the analysis of glycolytic enzymes related to carbohydrate metabolism in clinical and environmental strains of *C. neoformans*, identified using the dbCAN program with the HMMER profile (E-value < 1 × 10^−15^, coverage > 0.35) (Table 2). Only enzyme profiles with a minimum relative abundance difference of 10% between clinical and environmental subgroups were selected. Enzymes with higher abundance in clinical strains are listed in the upper section of the table, while those with higher abundance in environmental strains are listed in the lower section. This table summarizes only the results with significant differences, while the full table containing all analyzed enzyme profiles is available in a public database referenced in the Section 2 for detailed consultation.

The metabolites associated with glycolytic enzymes in *C. neoformans* were identified based on relative abundance in clinical and environmental strains. Following the determination of enzyme patterns for each subgroup, the corresponding amino acid residue sequences were extracted from NCBI and analyzed in the Human Metabolome Database (HMDB). Metabolite–enzyme interactions were categorized according to their higher abundance in each subgroup. Metabolites unique to each niche were utilized to explore potential metabolic pathway products, contributing to the understanding of specific adaptations within each subgroup and their possible role in virulence factors and antifungal resistance (see Appendix A).

The metabolic pathway enrichment analysis of *C. neoformans* identified significant differences between clinical and environmental isolates concerning the abundance of metabolites across specific pathways, as detailed in Table 3 and visualized in Figure 2A,B. To address the high number of comparisons made in this study and reduce the risk of false positives, the Benjamini–Hochberg procedure was applied to adjust *p*-values, resulting in false discovery rate (FDR)-corrected q-values. These adjustments provided a robust statistical foundation for interpreting the enriched pathways.

For clinical isolates, sulfur metabolism was the most significantly enriched pathway (*p* = 0.0053; FDR = 0.39271; impact = 0.0621). This finding suggests that these isolates adapt to oxidative stress, a hallmark of host environments, by employing sulfur metabolism for redox balance and detoxification. Other pathways, such as amino sugar and nucleotide sugar metabolism (*p* = 0.0237; FDR = 0.87731; impact = 0.1993) and cysteine and methionine metabolism (*p* = 0.0427; FDR = 0.97982; impact = 0.10096), support amino acid biosynthesis and antioxidant responses essential for survival in the host.

In environmental isolates, methane metabolism emerged as the most significant pathway (*p* = 0.000070346; FDR = 0.0052056; impact = 0.10417), reflecting adaptations to carbon-rich substrates prevalent in external environments. Glyoxylate and dicarboxylate metabolism (*p* = 0.0038; FDR = 0.14125) and taurine and hypotaurine metabolism (*p* = 0.0372; FDR = 0.91726) were also enriched, indicating metabolic strategies to manage diverse environmental stresses, including nutrient variability and oxidative conditions.

These findings, summarized in Table 3, are visualized in Figure 2. Panel (a) highlights the pathways with higher abundance in clinical isolates, while panel (b) illustrates pathways enriched in environmental isolates. The scatter plots demonstrate the relationship between the impact score (x-axis) and statistical significance (−log10(*p*), y-axis), with color gradients representing adjusted q-values and circle sizes indicating enrichment ratios. This comprehensive visualization reinforces the distinct metabolic adaptations between the two groups.

### 3.4. Metabolic Pathway Enrichment Analysis of C. neoformans Clinical and Environmental Isolates

The metabolic pathway enrichment analysis revealed distinct pathways enriched in *C. neoformans* strains isolated from clinical and environmental samples, as shown in Table 4 and illustrated in Figure 3A,B. To ensure statistical robustness and address the multiple comparison issue raised by the large dataset, *p*-values were adjusted using the Benjamini–Hochberg procedure to calculate the false discovery rate (FDR). This adjustment ensures that the reported differences are not only statistically significant but also reliable across multiple comparisons.

For clinical isolates, the most enriched pathway was glutathione oxidoreductase (*p* = 0.259; FDR = 0.0202), a critical pathway involved in the antioxidant response and cellular redox balance, suggesting a key adaptation to the oxidative stress encountered in host environments. Other enriched pathways included iodide oxidoreductase (*p* = 0.0647; FDR = 0.0635) and thyroid peroxidase (*p* = 0.0517; FDR = 0.0511), which also contribute to the enhanced antioxidant capacity. Pathways related to sulfur-containing amino acid metabolism, such as cystathionine beta-synthase (*p* = 0.0776; FDR = 0.0759), highlight a possible role in detoxification and redox regulation within clinical settings.

Environmental isolates showed enrichment in glutathione peroxidase (e) (*p* = 0.0172; FDR = 0.0172), a pathway linked to oxidative stress defense mechanisms in challenging environments. Additionally, pathways such as oxidized glutathione exchange (*p* = 0.0172; FDR = 0.0172) and methenyltetrahydrofolate cyclohydrolase, mitochondrial (*p* = 0.069; FDR = 0.0679) indicate metabolic strategies to counter environmental oxidative conditions. Other pathways, such as glycine hydroxymethyltransferase (*p* = 0.155; FDR = 0.149) and phosphoserine transaminase (*p* = 0.198; FDR = 0.189), emphasize the role of serine and folate metabolism in supporting survival under nutrient-limited conditions.

These results, summarized in Table 4, are visually represented in Figure 3. Panel (a) shows the enriched pathways in clinical isolates, while panel (b) illustrates pathways enriched in environmental isolates. The scatter plots highlight the relationship between pathway impact scores (x-axis) and −log10(*p*) values (y-axis). Circle sizes represent enrichment ratios, and color gradients reflect FDR-adjusted q-values, with darker colors indicating higher statistical significance. This visualization provides a clear distinction between the metabolic adaptations of clinical and environmental isolates of *C. neoformans*.

### 3.5. Comparative Analysis of Enzymatic Patterns and Virulence Profiles in C. neoformans Across Pathogenic Contexts

The results reveal a strong association between the enzymatic pattern of *C. neoformans* and various pathogens, as indicated by contingency analysis and the chi-square test. The contingency table used for this analysis, along with the detailed enzymatic frequencies, is available for consultation in the Open Science Framework (OSF) repository (https://doi.org/10.17605/OSF.IO/8JF2K) (accessed on 15 January 2025). The test yielded a chi-square value of 226.31 with a highly significant *p*-value of 6.79 × 10^−11^, suggesting a significant correlation between the enzymes present in *C. neoformans* and the analyzed pathogens. These findings point to the specificity of *C. neoformans* enzymes to certain pathogens, potentially playing an important role in virulence mechanisms and resistance.

Upon analysis of the contingency table, it was observed that the enzyme alpha-glucosidase was particularly prevalent in *Aspergillus fumigatus*, with five occurrences, a frequency not observed in other pathogens. However, it was also detected in one occurrence in *Fusarium oxysporum* and three in *Mycobacterium tuberculosis*, suggesting that this enzyme may play a role in specific interactions with these organisms. The enzyme N-acetylglucosamine-6-phosphate deacetylase was notably present in two samples of *Candida albicans* and one of *Magnaporthe oryzae*, which may be related to the reduced virulence observed in these pathogens. These data indicate that the enzymatic distribution varies considerably among different pathogens, reflecting evolutionary adaptations that may be associated with antimicrobial resistance and pathogenic capability.

The enzyme streptomycin biosynthesis protein StrI, in turn, appeared exclusively in *Legionella pneumophila*, a pathogen known for its antimicrobial resistance. This finding may suggest that the presence of this protein plays a critical role in resistance mechanisms associated with this specific pathogen. Furthermore, the high prevalence of alpha-glucosidase in *M. tuberculosis* may be linked to increased virulence in this pathogen, as evidenced by its frequency in the analyzed samples.

Figure 4A presents a heatmap showing the correlation patterns between *C. neoformans* enzymatic profiles and various pathogens. The heatmap reflects frequency distributions and association patterns, providing a visual representation of the relationships among the analyzed variables. While the chi-square test quantified the statistical significance of these associations, the heatmap serves as a complementary tool to explore trends and variability in the enzymatic data. The color gradient reflects the intensity of enzymatic presence in each pathogen, with darker colors indicating higher frequency. The visualization also highlights the concentration of specific enzymes in pathogens such as *A. fumigatus* and *Pseudomonas aeruginosa*, suggesting that enzymatic variability is influenced by pathogen type and possibly correlates with each one’s virulent capacity.

In addition, Figure 4B presents a comparative analysis of virulence profiles between clinical and environmental isolates of *C. neoformans*, revealing significant differentiation between the two groups. Clinical isolates exhibit a higher frequency of profiles associated with both unaffected pathogenicity and hypervirulence, suggesting that these isolates retain or even enhance their virulence potential in clinical settings. This observation does not imply an adaptive reduction in virulence within the human host; instead, it may reflect metabolic and genetic traits that incidentally contribute to increased virulence, rather than a direct selection for lower pathogenicity. In contrast, environmental isolates exhibit a broader range of virulence profiles, with a lower frequency of reduced virulence categories, which may be influenced by different selective pressures acting in their native habitats. These findings highlight distinct evolutionary dynamics between clinical and environmental strains of *C. neoformans*.

The hypervirulence category was not widely represented in either group but showed a slight predominance in clinical isolates. This finding may suggest a specific response by some strains to pressures from the hospital or human environment. The plant avirulence determinant profile was observed less frequently across isolates from both groups, indicating that *C. neoformans* shows a preferential adaptation to animal or human hosts rather than plant hosts.

These observations are supported by the color gradient in Figure 4B, which illustrates the frequency of each virulence category, with darker shades indicating higher prevalence. Differences between clinical and environmental isolates suggest that *C. neoformans* may adjust its virulence and pathogenicity depending on the ecological niche, possibly as a strategy to maximize its survival and infection potential in different contexts. Together, Figure 4A,B demonstrate that both the enzymatic and virulence profiles of *C. neoformans* are shaped by environmental and clinical factors, reflecting adaptations that may have direct implications for virulence and antimicrobial resistance.

## 4. Discussion

The phenetic and metabolomic adaptations in *C. neoformans*, particularly the differentiation between clinical and environmental isolates, provide valuable insights into the pathogen’s resilience. These adaptations underline its ability to navigate diverse ecological niches, offering mechanisms that enhance survival and virulence. Our findings support the hypothesis that *C. neoformans* adapts to selective pressures across diverse environments, resulting in distinct genetic and phenotypic traits that enhance its ability to survive in both clinical and environmental settings. This adaptability reinforces the role of niche-specific pressures in shaping *C. neoformans*’s evolutionary pathways, adding new insights into its virulence and antimicrobial resistance mechanisms [1,9,21].

The inclusion of other *Cryptococcus* species, such as *C. gattii*, was not pursued due to their distinct clinical and ecological profiles. By focusing exclusively on *C. neoformans*, this study aimed to elucidate specific phenetic and metabolomic adaptations without introducing confounding variables associated with interspecies comparisons. For instance, *C. gattii* is more commonly associated with infections in immunocompetent hosts, contrasting with *C. neoformans*, which predominantly affects immunosuppressed individuals [22,23]. Incorporating additional species would have diverged from the primary objective of exploring phenetic and metabolomic adaptations in *C. neoformans*.

A significant finding of this study was the distinct molecular phenetic clustering observed between clinical and environmental strains of *C. neoformans* var. *grubii.* This clustering highlights evolutionary pressures driving niche-specific adaptations and underscores the pathogen’s genetic and ecological plasticity. The formation of two main clades, with distinct subgroups for clinical and environmental isolates, suggests general trends of phenetic divergence influenced by their isolation source. This analysis includes the reference strain JEC21, which serves as an evolutionary baseline to contextualize the clustering patterns observed [24,25].

While the clades predominantly reflect clinical and environmental groupings, the overlap of some isolates, such as the two final strains in the tree belonging to both clinical and environmental groups, underscores the genetic and ecological plasticity of *C. neoformans*. These patterns may result from cross-adaptation events or shared evolutionary pressures that allow certain strains to transition between ecological niches [26,27].

Using the UPGMA method with 1000 bootstrap replicates, our phylogenetic analysis revealed the distinct clustering of clinical and environmental isolates, underscoring phenetic divergence. However, reliance on a single genetic marker, the multidrug efflux pump gene, may have limited the resolution, and broader genomic analyses are recommended to validate these evolutionary patterns. This phenomenon indicates a genetic and evolutionary divergence shaped by specific environmental factors in clinical and non-clinical settings, as observed in previous studies that examined molecular phenetic relationships in *Cryptococcus* species [9,28]. The separation of clinical isolates into distinct molecular phenetic groups may be reflective of selective pressures encountered within human hosts, where the pathogen must counter host immune defenses and adapt to intracellular environments rich in oxidative stress, as has been shown in studies emphasizing the metabolic and genetic plasticity of *C. neoformans* in response to host immunity [3,29].

Clinical strains of *C. neoformans* exhibit significant enrichment in sulfur metabolism- and glutathione-associated pathways, emphasizing their adaptation to oxidative stress in host environments [30,31]. This metabolic plasticity highlights a key survival strategy under immune pressures. To ensure the robustness of these findings, *p*-values were adjusted using the Benjamini–Hochberg procedure, addressing the issue of false discoveries across multiple comparisons and providing reliable insights into metabolic adaptations [17]. This adaptation aligns with findings from previous studies that identified sulfur-based metabolic pathways as vital for pathogen survival under oxidative stress conditions [10,21]. The antioxidant mechanisms in clinical isolates, particularly those involving sulfur metabolism and glutathione pathways, represent key virulence factors. These adaptations enable *C. neoformans* to withstand oxidative stress imposed by host immunity, sustaining infection and contributing to its pathogenicity [32,33].

Moreover, environmental isolates demonstrated unique adaptations that enable survival in carbon-rich external environments. The significant presence of methane metabolism and glyoxylate pathways in these isolates suggests that *C. neoformans* possesses specific metabolic traits to exploit organic compounds in soil and other environmental substrates [11,34]. Similarly to the clinical isolates, the statistical analyses for environmental isolates included FDR adjustments to ensure that the observed enrichments are both statistically and biologically meaningful, providing confidence in the distinct metabolic strategies identified. This is consistent with studies indicating that *C. neoformans* adapts its metabolism to nutrient availability in different ecological niches, thus enabling survival across a range of habitats [34,35,36]. Furthermore, although our study did not directly evaluate the effects of agricultural fungicides, such as triazoles, previous research has highlighted their role in inducing cross-resistance to clinical azoles. This underscores the need for vigilance in monitoring environmental reservoirs, where fungicide exposure may inadvertently select for resistant strains [10,33]. The observed clustering patterns and metabolic adaptations in environmental isolates suggest that metabolic flexibility may confer resilience to environmental pressures, including fungicide exposure. Previous studies have shown that agricultural fungicides, such as triazoles, can inadvertently induce cross-resistance to clinical antifungal agents, a phenomenon that warrants further investigation [10,33].

The identification of cross-adapted isolates, such as MW-RSA852, within clinical clades underscores significant public health implications. These findings highlight the potential for environmental strains to acquire virulence traits and transition into human pathogens, complicating clinical management and emphasizing the importance of monitoring environmental reservoirs. Similar cross-environmental adaptations have been documented in other studies, suggesting that *C. neoformans* may serve as a model for understanding zoonotic transmission and environmental influences on pathogen emergence [1,3,37]. This finding aligns with global epidemiological trends showing increasing incidences of cryptococcal infections derived from environmental sources, emphasizing the need for vigilant surveillance of environmental reservoirs of *C. neoformans* [11,21].

This study’s metabolomic profiling further distinguishes the unique metabolic strategies between clinical and environmental isolates. The identification of higher abundances of glycolytic enzymes, particularly those involved in sulfur and nitrogen metabolism in clinical strains, suggests a functional role in supporting intracellular survival by maintaining redox balance and nutrient assimilation under host-imposed stresses. Importantly, the application of the Benjamini–Hochberg correction ensured that the reported enriched pathways and enzymatic profiles are statistically robust, minimizing the likelihood of false positives and strengthening the validity of these observations. These adaptations reflect findings from recent studies that highlight glycolytic enzymes and sulfur-related pathways as targets of metabolic modulation in pathogenic *Cryptococcus* strains [9,28,38]. Conversely, environmental strains’ reliance on methanotrophic pathways for survival in organic matter-rich environments emphasizes the metabolic flexibility that enables *C. neoformans* to colonize a range of habitats and presents potential areas for targeted intervention strategies in environmental reservoirs [11,39].

The analysis of enzymatic patterns and virulence profiles revealed significant associations between the enzymatic profiles of *C. neoformans* and its adaptability and pathogenicity in various contexts. The correlation between the enzymes present and the observed pathogenicity highlights the relevance of specific enzymatic factors in virulence mechanisms and antimicrobial resistance. This association reinforces previous studies that identify enzymes as critical factors in fungal pathogenesis, including *C. neoformans* [40].

The prominence of the enzyme alpha-glucosidase, predominantly found in *Aspergillus fumigatus*, with additional occurrences in *Fusarium oxysporum* and *Mycobacterium tuberculosis*, suggests an adaptive role in interactions with specific pathogens. Previous studies have demonstrated that glucose-related enzymes can influence infection and dissemination processes in various microorganisms, indicating that the presence of this enzyme may be linked to enhanced survival in specific environments [41].

Another significant finding was the prevalence of the StrI protein, associated with streptomycin biosynthesis, exclusively in *Legionella pneumophila*. This enzyme may play a central role in antimicrobial resistance, a phenomenon widely documented in pathogens adapted to hospital environments [42]. The detection of the enzyme N-acetylglucosamine-6-phosphate deacetylase in *Candida albicans* and *Magnaporthe oryzae* suggests a reduction in virulence in these organisms, supporting the idea that alterations in enzymatic metabolism can directly affect infectious capacity [21].

Our use of the PHI-base database enabled the identification of putative virulence genes in *C. neoformans* isolates. While PHI-base is a robust tool for pathogenicity-related genes, it includes diverse pathogens with varying virulence strategies, some of which may not directly apply to *C. neoformans*. For example, genes associated with intracellular growth in macrophages, as seen in *Legionella pneumophila*, may not align with *C. neoformans*’s mechanisms of evading host defenses [43,44]. Despite this limitation, the database provides predictive insights that can guide experimental validation under host-relevant conditions. Future studies integrating genomic data with functional assays will be crucial to confirming the role of these genes in the pathogenesis of *C. neoformans*.

The virulence profiles, presented in the heatmap, revealed marked differences between clinical and environmental isolates of *C. neoformans*. Clinical isolates exhibited a higher frequency of profiles associated with unaffected pathogenicity, indicating an adaptation to the human environment that preserves infectious capacity even under environmental challenges. Conversely, environmental isolates showed a lower rate of reduced virulence, reflecting distinct selective pressures likely associated with survival in environments that are nutrient-rich but less prone to immune challenges [45].

The subtle predominance of the hypervirulence category in clinical isolates suggests specific responses to hospital-related pressures, aligning with studies showing that virulence factors can be modulated by environmental conditions such as pH, nutrient availability, and quorum sensing signals [21,41]. The role of glutathione in redox regulation, demonstrated by Black et al. (2024) [45], may also contribute to these adaptations, providing a crucial metabolic advantage for countering host oxidative stress.

Finally, the lower number of virulence profiles associated with plant avirulence determinants highlights the specialization of *C. neoformans* in animal and human hosts. Recent studies reinforce that the environment plays a critical role in regulating virulence factors, modulating not only gene expression but also susceptibility to antifungal therapies [41].

These findings, contextualized with the literature, underscore the need for complementary experimental investigations to validate the mechanisms observed in silico. Future studies could delve deeper into the interactions between enzymatic profiles and virulence determinants, elucidating the molecular bases guiding *C. neoformans* adaptation to diverse ecological niches and clinical contexts.

This study provides important insights into the phenetic and metabolomic adaptations of *C. neoformans*; however, certain limitations should be acknowledged. While this study leveraged an in silico approach to identify genomic and metabolic trends, the absence of direct experimental validation limits the ability to fully capture the complex interplay between environmental and host-specific pressures on *C. neoformans* adaptation. Nonetheless, statistical corrections, such as false discovery rate adjustments, ensured the reliability and interpretability of the findings across a large dataset. Experimental validation through in vitro and in vivo models would significantly strengthen these findings, offering a more direct view of phenotypic plasticity and adaptive mechanisms under controlled stress conditions [28,29].

Although this study included isolates from diverse geographic regions, the limited sample size within each category may restrict the generalizability of our findings. Expanding the dataset in future research could provide deeper insights into regional adaptations and enhance the robustness of observed trends. Although the trends observed are meaningful and supported by statistical analyses, expanding the dataset to include a broader and more globally representative range of isolates would provide deeper insights into regional adaptations and cross-resistance patterns. This would also mitigate potential biases arising from small sample sizes and enhance the robustness of the findings [3,9].

A notable limitation of this study is the temporal aspect of the isolates included. Some isolates were collected over two decades ago; however, it is important to emphasize that their sequences were deposited more recently in the NCBI database, reflecting advancements in sequencing technology and data sharing practices. These deposited sequences remain relevant for studying core mechanisms of *C. neoformans* pathogenicity and adaptation, as traits such as capsule formation, melanin synthesis, and sulfur metabolism are evolutionarily conserved. Additionally, newer genomic datasets with comprehensive annotations are often not immediately available on open access platforms. It is anticipated that future studies will benefit from more recent isolates as these data are deposited and shared in public repositories, enabling a broader temporal perspective on emerging adaptations and resistance mechanisms.

This study primarily focused on glycolytic and sulfur-related pathways due to their established associations with virulence and oxidative stress resistance. However, other metabolic pathways potentially influencing these traits, such as lipid metabolism, amino acid biosynthesis, and secondary metabolite production, were not fully explored. Future metabolomic analyses encompassing these pathways could provide a more comprehensive understanding of the biochemical adaptations that underlie virulence and antifungal resistance in *C. neoformans*.

Lastly, the observed cross-adaptation event involving the C45 isolate requires further investigation. This clinical isolate’s unexpected clustering with environmental strains suggests the presence of transitional mechanisms that may enable environmental strains to acquire pathogenic traits. To better elucidate these mechanisms, future studies should integrate genomic, metabolomic, and phenetic analyses, complemented by functional experiments. Such an approach would offer critical insights into the genetic and phenotypic factors driving environmental-to-clinical transitions and highlight the evolutionary pressures shaping *C. neoformans* adaptability, with implications for public health [10,33].

## 5. Conclusions

This study highlights significant phenetic and metabolomic distinctions between clinical and environmental isolates of *C. neoformans*, shedding light on the adaptive mechanisms that may contribute to virulence and potentially antifungal resistance across distinct ecological contexts. The molecular phenetic separation observed between clinical and environmental isolates underscores niche-specific evolutionary pressures that shape the phenetic diversity within this pathogen. Metabolomic profiling revealed that clinical isolates exhibit enriched sulfur metabolism- and glutathione-related pathways, adaptations likely crucial for oxidative stress resistance in host environments. In contrast, environmental isolates demonstrate enhanced pathways for methane metabolism and glyoxylate utilization, suggesting adaptations for survival in carbon-rich, nutrient-variable environments.

The identification of enzymes and metabolic pathways with distinct frequencies in clinical and environmental isolates, as presented in our results, highlights potential mechanisms that could indirectly influence antifungal resistance. While this study did not directly assess antifungal resistance, the findings provide valuable insights into the metabolic plasticity of *C. neoformans* that might contribute to its ability to tolerate antifungal stress, as suggested by the previous literature.

The presence of cross-adapted isolates within clinical clades suggests potential environmental origins or cross-adaptation events that could complicate clinical management. These findings emphasize the importance of monitoring environmental reservoirs for pathogenic variants of *C. neoformans* and support the need for targeted therapeutic strategies that consider the pathogen’s metabolic versatility.

Future research should validate these in silico findings through experimental approaches and expand the dataset to capture a wider geographic and ecological diversity of isolates. Such studies could enhance our understanding of *C. neoformans*’s resilience, including its potential to develop antifungal resistance, and inform strategies for managing cryptococcosis, especially given the pathogen’s ability to adapt to both environmental and host-specific stressors.

## Figures and Tables

**Figure 1 jof-11-00215-f001:**
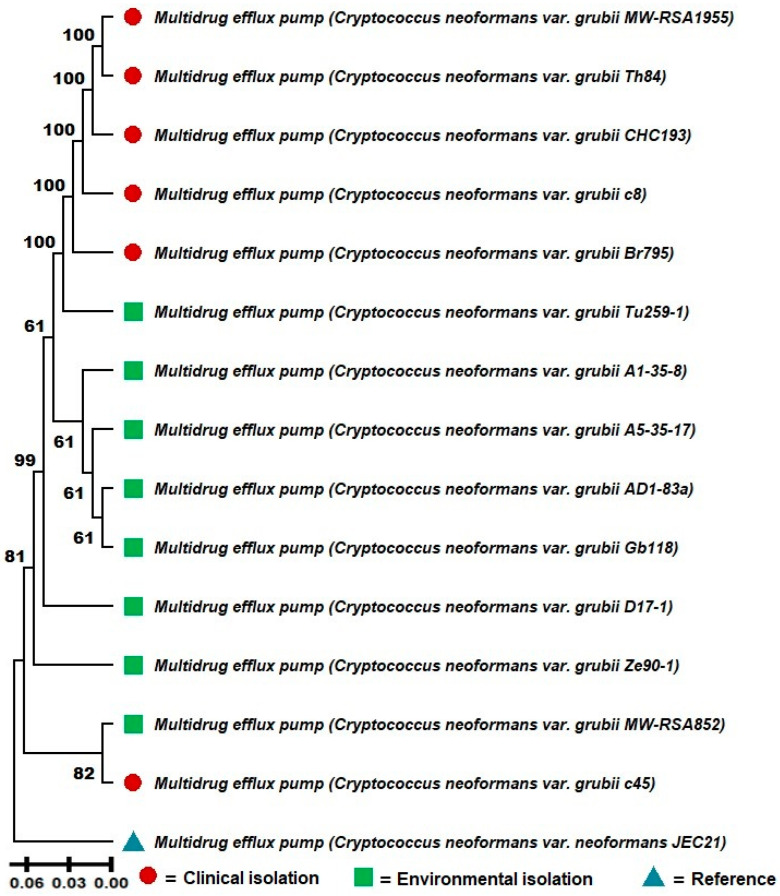
Molecular phenetic tree of *C. neoformans* var. *grubii*. The molecular phenetic tree of *C. neoformans* var. *grubii* was constructed based on multidrug efflux pump protein sequences. The tree was generated using the UPGMA clustering method with 1000 bootstrap replicates and the Dayhoff evolutionary model. Clinical and environmental samples form distinct clades with strong statistical support, reflecting potential phenetic differentiation between these populations. Notably, clinical isolate C45 clusters with environmental samples, which may indicate a shared evolutionary mechanism, such as efflux pump adaptation, enabling survival in diverse environments. The reference strain JEC21 was included to provide a baseline comparison and is positioned outside the clinical and environmental clades. The VNII isolate was excluded from this analysis as the multidrug efflux pump protein sequence was not available. The red circle denotes the clinical isolates, the green square highlights the environmental isolates, and the blue triangle identifies the reference isolate JEC21.

**Figure 2 jof-11-00215-f002:**
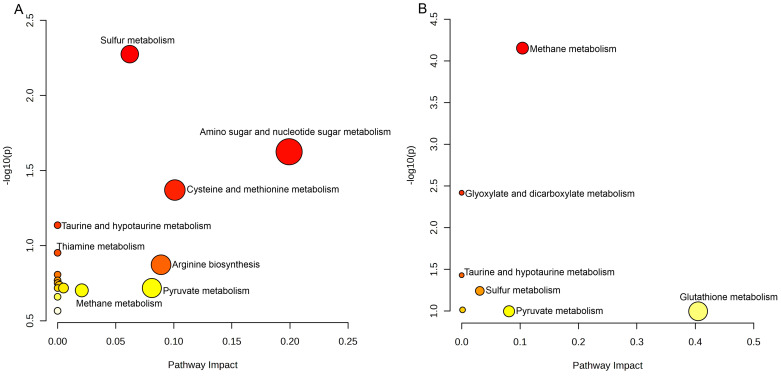
Distribution of major metabolic pathways in *C. neoformans* isolates. (**A**) Scatter plot displaying the primary metabolic pathways with higher abundance in clinical isolates. (**B**) Scatter plot illustrating the main pathways enriched in environmental isolates. The x-axis represents the pathway impact score, while the y-axis shows the −log10(*p*) values. Adjusted *p*-values (FDR) were computed using the Benjamini–Hochberg procedure to account for multiple comparisons. Circle sizes indicate enrichment ratios, with larger circles denoting higher enrichment. Color gradients reflect q-values, with darker colors representing greater statistical significance. This figure underscores the metabolic distinctions between clinical and environmental isolates, suggesting adaptations to oxidative stress in clinical isolates and carbon-rich substrates in environmental isolates.

**Figure 3 jof-11-00215-f003:**
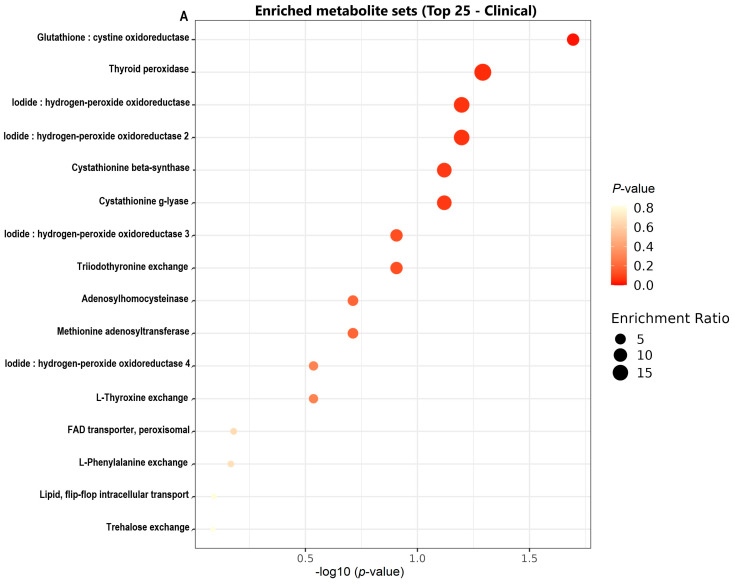
Distribution of enriched metabolite sets in *C neoformans* strains with higher abundance in clinical (**A**) and environmental (**B**) isolates. (**A**) Scatter plot showing enriched metabolic pathways in clinical isolates. (**B**) Scatter plot showing enriched pathways in environmental isolates. The x-axis represents the pathway impact score, while the y-axis shows −log10(*p*). False discovery rate (FDR)-adjusted q-values were calculated using the Benjamini–Hochberg procedure to account for multiple comparisons. Circle sizes denote enrichment ratios, with larger circles indicating greater enrichment. The color gradient reflects q-values, with darker colors corresponding to higher significance. This figure highlights the distinct metabolic adaptations of clinical and environmental isolates.

**Figure 4 jof-11-00215-f004:**
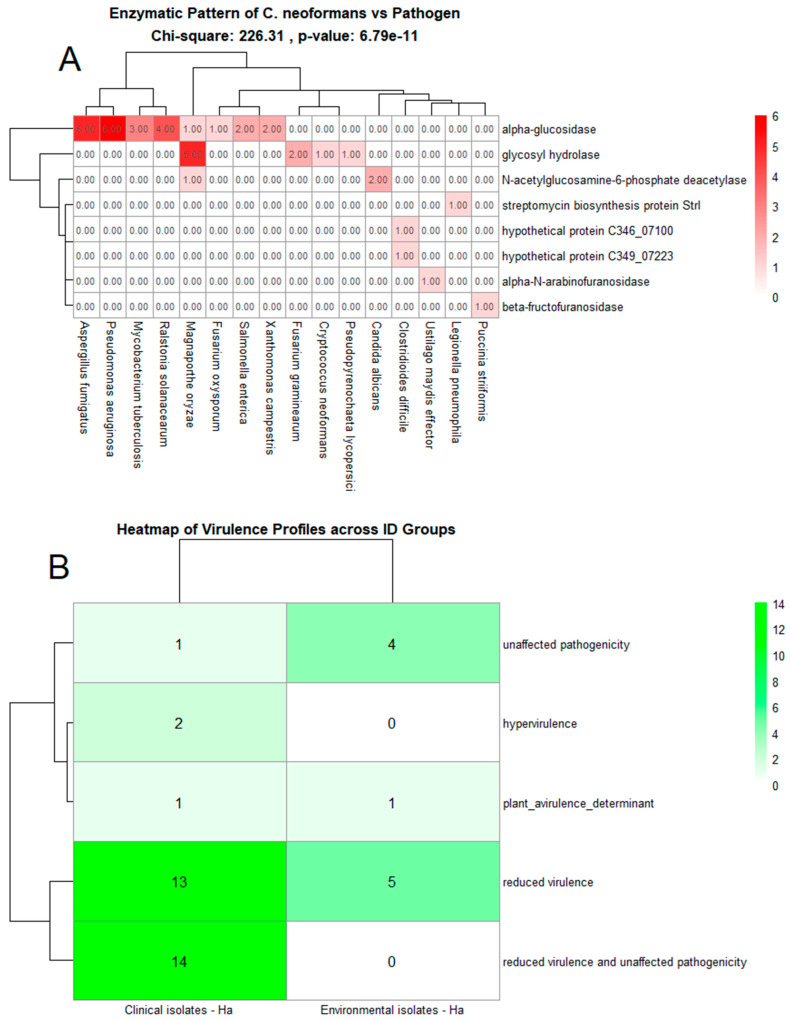
Virulence and enzymatic profile analysis of *C. neoformans* strains in comparison with various pathogens. (**A**) Enzymatic pattern comparison between *C. neoformans* and a range of pathogenic species. This heatmap represents the presence and abundance of specific enzymes across the tested organisms, with a particular focus on *C. neoformans*. Enzymes such as alpha-glucosidase, glycosyl hydrolase, and N-acetylglucosamine-6-phosphate deacetylase are shown, highlighting enzymatic activity across different pathogens. The color gradient from light to dark red reflects the frequency of each enzyme, with darker colors indicating higher abundance. The chi-square value (226.31) and *p*-value (6.79 × 10^−11^) demonstrate statistically significant differences in enzymatic profiles across the species. (**B**) Heatmap of virulence profiles across clinical and environmental *C. neoformans* isolates. This heatmap illustrates different virulence categories identified through PHI-base, including unaffected pathogenicity, hypervirulence, plant avirulence determinants, reduced virulence, and reduced virulence with unaffected pathogenicity. While these categories provide predictive insights into potential virulence strategies, they should be interpreted cautiously, as some virulence factors may originate from pathogens with distinct infection mechanisms. Experimental validation is necessary to confirm their relevance to *C. neoformans*.

**Table 1 jof-11-00215-t001:** Characteristics of *C. neoformans* strains selected from NCBI database for phenetic analysis and enzymatic pattern exploration.

Strain	Taxon	Collection Date	Information	Geo Loc Name	Isolation Source	Host
JEC21	*Cryptococcus neoformans* var. *neoformans*	7 January 2005	Reference	-	-	-
VNII	*Cryptococcus neoformans*	2000	Environmental	USA	Cockatoo excrement	Missing
MW-RSA852	*Cryptococcus neoformans* var. *grubii*	17 July 2017	Environmental	South Africa	Mopane	Mopane
c45	*Cryptococcus neoformans* var. *grubii*	2001	Clinical	USA	Sputum	Human
A1-35-8	*Cryptococcus neoformans* var. *grubii*	Missing	Environmental	USA	Pigeon guano	Pigeon
AD1-83a	*Cryptococcus neoformans* var. *grubii*	Missing	Environmental	France	Missing	Mopane
CHC193	*Cryptococcus neoformans* var. *grubii*	1998	Clinical	China	Cerebrospinal fluid	Human
A5-35-17	*Cryptococcus neoformans* var. *grubii*	Missing	Environmental	USA	Pigeon guano	Eucalyptus
MW-RSA1955	*Cryptococcus neoformans* var. *grubii*	Missing	Clinical	South Africa	Clinical	Human
Br795	*Cryptococcus neoformans* var. *grubii*	2006	Clinical	Brazil	Clinical	Human
D17-1	*Cryptococcus neoformans* var. *grubii*	Missing	Environmental	South Africa	Missing	Pigeon
Ze90-1	*Cryptococcus neoformans* var. *grubii*	Missing	Environmental	South Africa	Eucalyptus tree	Human
c8	*Cryptococcus neoformans* var. *grubii*	2000	Clinical	USA	Cerebrospinal fluid	Human
Th84	*Cryptococcus neoformans* var. *grubii*	2006	Clinical	Thailand	Blood	Human
Tu259-1	*Cryptococcus neoformans* var. *grubii*	Missing	Environmental	Botswana	Mopane tree	Human
Gb118	*Cryptococcus neoformans* var. *grubii*	2007	Environmental	Botswana	Guano	Pigeon

**Table 2 jof-11-00215-t002:** Abundance of glycolytic enzymes related to carbohydrate metabolism in clinical and environmental strains of *C. neoformans* (dbCAN analysis, HMMER: E-value < 1 × 10^−15^, coverage > 0.35, ≥10%).

HMM Profile	Query ID	Clinical Sample—%	Environmental—%	%—Difference
Higher Abundance in Clinical Strains				
CBM91.hmm	OXG74472.1	0.081	0.055	47.27
CE9.hmm	OXG11898.1	0.568	0.492	15.45
GH109.hmm	OXG11896.1	0.568	0.492	15.45
GH13_30.hmm	OXG22770.1	0.487	0.437	11.44
GH13_40.hmm	OXG41196.1	0.893	0.656	36.13
GH154.hmm	OXG15174.1	0.487	0.437	11.44
GH177.hmm	UOH82297.1	3.896	3.499	11.35
GH43_13.hmm	OXG71282.1	0.081	0.055	47.27
GH51_1.hmm	UOH84770.1	0.487	0.437	11.44
Higher Abundance in Environmental Strains				
AA9.hmm	OXG28961.1	0.487	0.547	12.32
CE1.hmm	OXC58165.1	0.893	0.984	10.19
GH32.hmm	OWZ48554.1	0.325	0.383	17.85

**Table 3 jof-11-00215-t003:** Metabolic pathways associated with glycolytic enzymes showing higher abundance in *C. neoformans* strains isolated from clinical and environmental samples.

Pathway Name	*p*	−log(*p*)	Holm *p*	Adjusted *p*-Value (FDR)	Impact
Higher Abundance in Strains Isolated from Clinical Samples					
Sulfur metabolism	0.0053069	2.2752	0.39271	0.39271	0.06214
Amino sugar and nucleotide sugar metabolism	0.023711	1.625	1.0	0.87731	0.1993
Cysteine and methionine metabolism	0.042662	1.37	1.0	0.97982	0.10096
Taurine and hypotaurine metabolism	0.073138	1.1359	1.0	0.97982	0.0
Thiamine metabolism	0.11171	0.9519	1.0	0.97982	0.0
Arginine biosynthesis	0.13418	0.87233	1.0	0.97982	0.08906
Valine, leucine and isoleucine biosynthesis	0.15614	0.80649	1.0	0.97982	0.0
Alanine, aspartate and glutamate metabolism	0.17051	0.76825	1.0	0.97982	0.0
Tyrosine metabolism	0.17051	0.76825	1.0	0.97982	0.0
Pantothenate and CoA biosynthesis	0.17762	0.75052	1.0	0.97982	0.0
Glycolysis/Gluconeogenesis	0.18467	0.73361	1.0	0.97982	0.00153
Glyoxylate and dicarboxylate metabolism	0.19167	0.71746	1.0	0.97982	0.0
Glutathione metabolism	0.19167	0.71746	1.0	0.97982	0.00511
Pyruvate metabolism	0.19167	0.71746	1.0	0.97982	0.08115
Methane metabolism	0.19861	0.70199	1.0	0.97982	0.02083
Arginine and proline metabolism	0.21914	0.65928	1.0	1.0	0.0
Glycine, serine and threonine metabolism	0.27164	0.56601	1.0	1.0	0.0
Valine, leucine and isoleucine degradation	0.27164	0.56601	1.0	1.0	0.0
Higher Abundance in Strains Isolated from Environmental Samples					
Methane metabolism	0.000070346	4.1528	0.0052056	0.0052056	0.10417
Glyoxylate and dicarboxylate metabolism	0.0038175	2.4182	0.27868	0.14125	0.0
Taurine and hypotaurine metabolism	0.037186	1.4296	1.0	0.91726	0.0
Sulfur metabolism	0.057391	1.2412	1.0	1.0	0.03107
Glycolysis/Gluconeogenesis	0.096846	1.0139	1.0	1.0	0.00153
Pyruvate metabolism	0.10072	0.99687	1.0	1.0	0.08115
Glutathione metabolism	0.10072	0.99687	1.0	1.0	0.40491

**Table 4 jof-11-00215-t004:** Enrichment analysis of metabolite sets with higher abundance in *C. neoformans* strains isolated from clinical and environmental samples.

Enrichment Analysis—Higher Abundance in Strains Isolated from Clinical Samples				
**Metabolite Set**	**Total**	**Hits**	**Expect**	**Adjusted *p*-value (FDR)**
Glutathione:cystine oxidoreductase	20	2	0.259	0.0202
Thyroid peroxidase	4	1	0.0517	0.0511
Iodide:hydrogen-peroxide oxidoreductase	5	1	0.0647	0.0635
Iodide:hydrogen-peroxide oxidoreductase 2	5	1	0.0647	0.0635
Cystathionine beta-synthase	6	1	0.0776	0.0759
Cystathionine g-lyase	6	1	0.0776	0.0759
Iodide:hydrogen-peroxide oxidoreductase 3	10	1	0.129	0.124
Triiodothyronine exchange	10	1	0.129	0.124
Adenosylhomocysteinase	16	1	0.207	0.194
Methionine adenosyltransferase	16	1	0.207	0.194
Iodide:hydrogen-peroxide oxidoreductase 4	25	1	0.323	0.291
L-Thyroxine exchange	25	1	0.323	0.291
FAD transporter, peroxisomal	70	1	0.905	0.661
L-Phenylalanine exchange	73	1	0.944	0.68
Lipid, flip-flop intracellular transport	98	1	1.27	0.809
Trehalose exchange	100	1	1.29	0.818
Enrichment Analysis—Higher Abundance in Strains Isolated from Environmental Samples				
**Metabolite Set**	**Total**	**Hits**	**Expect**	** *p* ** **-value**
Glutathione peroxidase (e)	2	1	0.0172	0.0172
Oxidized glutathione exchange	2	1	0.0172	0.0172
Methenyltetrahydrifikate cyclohydrolase, mitochondrial	8	1	0.069	0.0679
Glycine hydroxymethyltransferase, reversible	18	1	0.155	0.149
Reduced glutathione exchange	18	1	0.155	0.149
Phosphoglycerate dehydrogenase	23	1	0.198	0.189
Phosphoserine phosphatase (L-serine)	23	1	0.198	0.189
Phosphoserine transaminase	23	1	0.198	0.189
FAD transporter, peroxisomal	70	1	0.603	0.513
L-Phenylalanine exchange	73	1	0.629	0.531
Lipid, flip-flop intracellular transport	98	1	0.845	0.667
Trehalose exchange	100	1	0.862	0.677

## Data Availability

The datasets generated and analyzed during the current study, including the DBCAN output, abundance ratio (%), and virulence analyses, are openly available in the Open Science Framework (OSF) repository at https://doi.org/10.17605/OSF.IO/8JF2K (accessed on 15 January 2025).

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
