# Peer review of "Adaptative Divergence of Cryptococcus neoformans: Phenetic and Metabolomic Profiles Reveal Distinct Pathways of Virulence and Resistance in Clinical vs. Environmental Isolates"

_jof, 2025, doi:10.3390/jof11030215_

Round 1
Reviewer 1 Report
This manuscript submitted by Camila Botelho Miguel and collaborators describes significant metabolomic distinctions among Cryptococcus isolates reveal important adaptive strategies that enhance virulence and antifungal resistance, highlighting potential therapeutic targets
Cryptococcus neoformans is a life-threatening fungal pathogen that primarily affects immunocompromised individuals. Although antiretroviral therapy has reduced its incidence in developed nations, fluconazole-resistant strains and virulent environmental isolates continue to pose challenges, especially because they have many adaptive mechanisms supporting their survival. This study explores the phenetic and metabolomic adaptations of C. neoformans in clinical and environmental contexts to understand the factors that influence pathogenicity and resistance.
This scientific work is characterized by being an in silico observational study conducted with 16 C. neoformans isolates (6 pathogenic, 9 environmental) from the NCBI database. Molecular phenetic analysis used the MEGA 11.0.13 tool and focused on efflux pump protein sequences. Phylogenetic relationships were assessed via Neighbor-Joining with 1,000 bootstrap replicates.
Enzyme profiling of glycolytic pathways was conducted with dbCAN, and metabolomic pathway enrichment analysis was performed in MetaboAnalyst 6.0 using the KEGG pathway database.
The results obtained with the phylogenetic analysis showed distinct clades for different isolates, indicating niche-specific phenetic divergence. Clinical isolates demonstrated enriched sulfur metabolism and glutathione pathways, likely adaptations to oxidative stress in host environments, while environmental isolates favored methane and glyoxylate pathways, suggesting adaptations for survival in carbon-rich environments.
The authors concluded that significant phenetic and metabolomic distinctions between isolates reveal adaptive strategies that increase virulence and antifungal resistance, highlighting potential therapeutic targets.
The manuscript is interesting, but since it is a work that describes the opportunistic fungus Cryptococcus neoformans, I missed a more detailed description of the polysaccharide capsule in the introduction of the manuscript. The authors could briefly talk about the polysaccharide components and some immunomodulatory effects. Below are some examples that could be used:
a) Decote-Ricardo D, LaRocque-de-Freitas IF, Rocha JDB, Nascimento DO, Nunes MP, Morrot A, Freire-de-Lima L, Previato JO, Mendonça-Previato L, Freire-de-Lima CG. Immunomodulatory Role of Capsular Polysaccharides Constituents of Cryptococcus neoformans. Front Med (Lausanne). 2019 Jun 19;6:129. doi: 10.3389/fmed.2019.00129.
b) Cryptococcus: History, Epidemiology and Immune Evasion
by Israel Diniz-Lima,Leonardo Marques da Fonseca,Elias Barbosa da Silva-Junior,Joyce Cristina Guimarães-de-Oliveira,Leonardo Freire-de-Lima,Danielle Oliveira Nascimento,Alexandre Morrot,Jose Osvaldo Previato,Lucia Mendonça-Previato,Debora Decote-Ricardo andCelio Geraldo Freire-de-Lima. Appl. Sci. 2022, 12(14), 7086; https://doi.org/10.3390/app12147086 -
c) Villena SN, Pinheiro RO, Pinheiro CS, Nunes MP, Takiya CM, DosReis GA, Previato JO, Mendonça-Previato L, Freire-de-Lima CG. Capsular polysaccharides galactoxylomannan and glucuronoxylomannan from Cryptococcus neoformans induce macrophage apoptosis mediated by Fas ligand. Cell Microbiol. 2008 Jun;10(6):1274-85. doi: 10.1111/j.1462-5822.2008.01125.x.
This manuscript submitted by Camila Botelho Miguel and collaborators describes significant metabolomic distinctions among Cryptococcus isolates reveal important adaptive strategies that enhance virulence and antifungal resistance, highlighting potential therapeutic targets
Cryptococcus neoformans is a life-threatening fungal pathogen that primarily affects immunocompromised individuals. Although antiretroviral therapy has reduced its incidence in developed nations, fluconazole-resistant strains and virulent environmental isolates continue to pose challenges, especially because they have many adaptive mechanisms supporting their survival. This study explores the phenetic and metabolomic adaptations of C. neoformans in clinical and environmental contexts to understand the factors that influence pathogenicity and resistance.
This scientific work is characterized by being an in silico observational study conducted with 16 C. neoformans isolates (6 pathogenic, 9 environmental) from the NCBI database. Molecular phenetic analysis used the MEGA 11.0.13 tool and focused on efflux pump protein sequences. Phylogenetic relationships were assessed via Neighbor-Joining with 1,000 bootstrap replicates.
Enzyme profiling of glycolytic pathways was conducted with dbCAN, and metabolomic pathway enrichment analysis was performed in MetaboAnalyst 6.0 using the KEGG pathway database.
The results obtained with the phylogenetic analysis showed distinct clades for different isolates, indicating niche-specific phenetic divergence. Clinical isolates demonstrated enriched sulfur metabolism and glutathione pathways, likely adaptations to oxidative stress in host environments, while environmental isolates favored methane and glyoxylate pathways, suggesting adaptations for survival in carbon-rich environments.
The authors concluded that significant phenetic and metabolomic distinctions between isolates reveal adaptive strategies that increase virulence and antifungal resistance, highlighting potential therapeutic targets.
The manuscript is interesting, but since it is a work that describes the opportunistic fungus Cryptococcus neoformans, I missed a more detailed description of the polysaccharide capsule in the introduction of the manuscript. The authors could briefly talk about the polysaccharide components and some immunomodulatory effects. Below are some examples that could be used:
a) Decote-Ricardo D, LaRocque-de-Freitas IF, Rocha JDB, Nascimento DO, Nunes MP, Morrot A, Freire-de-Lima L, Previato JO, Mendonça-Previato L, Freire-de-Lima CG. Immunomodulatory Role of Capsular Polysaccharides Constituents of Cryptococcus neoformans. Front Med (Lausanne). 2019 Jun 19;6:129. doi: 10.3389/fmed.2019.00129.
b) Cryptococcus: History, Epidemiology and Immune Evasion
by Israel Diniz-Lima,Leonardo Marques da Fonseca,Elias Barbosa da Silva-Junior,Joyce Cristina Guimarães-de-Oliveira,Leonardo Freire-de-Lima,Danielle Oliveira Nascimento,Alexandre Morrot,Jose Osvaldo Previato,Lucia Mendonça-Previato,Debora Decote-Ricardo andCelio Geraldo Freire-de-Lima. Appl. Sci. 2022, 12(14), 7086; https://doi.org/10.3390/app12147086 -
c) Villena SN, Pinheiro RO, Pinheiro CS, Nunes MP, Takiya CM, DosReis GA, Previato JO, Mendonça-Previato L, Freire-de-Lima CG. Capsular polysaccharides galactoxylomannan and glucuronoxylomannan from Cryptococcus neoformans induce macrophage apoptosis mediated by Fas ligand. Cell Microbiol. 2008 Jun;10(6):1274-85. doi: 10.1111/j.1462-5822.2008.01125.x.
Author Response
This manuscript submitted by Camila Botelho Miguel and collaborators describes significant metabolomic distinctions among Cryptococcus isolates reveal important adaptive strategies that enhance virulence and antifungal resistance, highlighting potential therapeutic targets Cryptococcus neoformans is a life-threatening fungal pathogen that primarily affects immunocompromised individuals. Although antiretroviral therapy has reduced its incidence in developed nations, fluconazole-resistant strains and virulent environmental isolates continue to pose challenges, especially because they have many adaptive mechanisms supporting their survival. This study explores the phenetic and metabolomic adaptations of C. neoformans in clinical and environmental contexts to understand the factors that influence pathogenicity and resistance.
This scientific work is characterized by being an in silico observational study conducted with 16 C. neoformans isolates (6 pathogenic, 9 environmental) from the NCBI database. Molecular phenetic analysis used the MEGA 11.0.13 tool and focused on efflux pump protein sequences. Phylogenetic relationships were assessed via Neighbor-Joining with 1,000 bootstrap replicates.
Enzyme profiling of glycolytic pathways was conducted with dbCAN, and metabolomic pathway enrichment analysis was performed in MetaboAnalyst 6.0 using the KEGG pathway database.
The results obtained with the phylogenetic analysis showed distinct clades for different isolates, indicating niche-specific phenetic divergence. Clinical isolates demonstrated enriched sulfur metabolism and glutathione pathways, likely adaptations to oxidative stress in host environments, while environmental isolates favored methane and glyoxylate pathways, suggesting adaptations for survival in carbon-rich environments.
The authors concluded that significant phenetic and metabolomic distinctions between isolates reveal adaptive strategies that increase virulence and antifungal resistance, highlighting potential therapeutic targets.
The manuscript is interesting, but since it is a work that describes the opportunistic fungus Cryptococcus neoformans, I missed a more detailed description of the polysaccharide capsule in the introduction of the manuscript. The authors could briefly talk about the polysaccharide components and some immunomodulatory effects. Below are some examples that could be used:
- a) Decote-Ricardo D, LaRocque-de-Freitas IF, Rocha JDB, Nascimento DO, Nunes MP, Morrot A, Freire-de-Lima L, Previato JO, Mendonça-Previato L, Freire-de-Lima CG. Immunomodulatory Role of Capsular Polysaccharides Constituents of Cryptococcus neoformans. Front Med (Lausanne). 2019 Jun 19;6:129. doi: 10.3389/fmed.2019.00129.
- b) Cryptococcus: History, Epidemiology and Immune Evasion by Israel Diniz-Lima,Leonardo Marques da Fonseca,Elias Barbosa da Silva-Junior,Joyce Cristina Guimarães-de-Oliveira,Leonardo Freire-de-Lima,Danielle Oliveira Nascimento,Alexandre Morrot,Jose Osvaldo Previato,Lucia Mendonça-Previato,Debora Decote-Ricardo andCelio Geraldo Freire-de-Lima. Appl. Sci. 2022, 12(14), 7086; https://doi.org/10.3390/app12147086 -
- c) Villena SN, Pinheiro RO, Pinheiro CS, Nunes MP, Takiya CM, DosReis GA, Previato JO, Mendonça-Previato L, Freire-de-Lima CG. Capsular polysaccharides galactoxylomannan and glucuronoxylomannan from Cryptococcus neoformans induce macrophage apoptosis mediated by Fas ligand. Cell Microbiol. 2008 Jun;10(6):1274-85. doi: 10.1111/j.1462-5822.2008.01125.x.
Answer: Dear Reviewer, we sincerely appreciate your detailed feedback and valuable suggestions to improve our manuscript. We acknowledge the importance of providing a more comprehensive description of the Cryptococcus neoformans polysaccharide capsule and its immunomodulatory effects, as highlighted in your review.
In response to your request, we have revised the introduction section to include detailed information about the primary components of the capsule, glucuronoxylomannan (GXM) and galactoxylomannan (GalXM), along with their immunomodulatory roles. Specifically, we have discussed the mechanisms by which GXM inhibits phagocytosis and modulates cytokine production, as well as how GalXM induces macrophage apoptosis via the Fas/FasL pathway, thereby subverting host immune responses. Furthermore, we highlighted the clinical relevance of these properties in promoting the persistence of the pathogen within host tissues and enabling systemic dissemination.
These additions were supported by the references you kindly suggested: Decote-Ricardo et al. (2019), Diniz-Lima et al. (2022), and Villena et al. (2008), which have been appropriately cited in the revised manuscript. For your convenience, the new text has been highlighted in yellow in the updated version of the manuscript.
Thank you once again for your insightful comments, which have undoubtedly strengthened our work. Should you have any additional suggestions or require further clarifications, please do not hesitate to reach out.

Reviewer 2 Report
The manuscript "Adaptative Divergence of Cryptococcus neoformans: Phenetic and Metabolomic Profiles Reveal Distinct Pathways of Viru-lence and Resistance in Clinical vs. Environmental Isolates" is representative of the kind of research in which open source database material becomes the subject of study. This kind of research is important because it provides a unified context for disparate findings and allows us to identify interrelationships based on the direction of thought of the authors. I believe that the manuscript in its present form can be published.
With the exception of the font in Figure 2 being too small, making it difficult to read, I found no major flaws in the manuscript materials.
Author Response
The manuscript "Adaptative Divergence of Cryptococcus neoformans: Phenetic and Metabolomic Profiles Reveal Distinct Pathways of Viru-lence and Resistance in Clinical vs. Environmental Isolates" is representative of the kind of research in which open source database material becomes the subject of study. This kind of research is important because it provides a unified context for disparate findings and allows us to identify interrelationships based on the direction of thought of the authors. I believe that the manuscript in its present form can be published.
Detail comments
With the exception of the font in Figure 2 being too small, making it difficult to read, I found no major flaws in the manuscript materials.
Answer: Dear Reviewer, we sincerely thank you for your thoughtful evaluation of our manuscript and for recognizing the importance of our work in utilizing open-source database materials to advance the understanding of Cryptococcus neoformans adaptations. Your positive feedback and support for the publication of our study are truly appreciated.
Regarding your specific comment on Figure 2, we fully agree that the font size in the original version was too small, which could hinder readability. We have addressed this issue by increasing the font size to ensure that all labels and elements in Figure 2 are now clear and legible. This adjustment is reflected in the updated manuscript.
Once again, we are grateful for your constructive feedback and kind words, which have greatly contributed to enhancing the clarity and impact of our work.

Reviewer 3 Report
This paper from Camila Botelho Miguel and collogues compares the genomes of a relatively small number of C. neoformans var Grubi from the NCBI database, after dividing then into pathogens (isolated from an infected host) or non-pathogens (environmental isolates). The entire paper is based on genomic analysis looking for pathways that were more highly expressed in one group or the other.
I found the paper confusing and hard to follow, starting with section 2.2 that lists 15 isolates and says they studied 16 isolates. While I agree that one can assume the clinical isolates are pathogenic, one cannot assume that the environmental isolates are non-pathogenic since infection occurs by inhalation of environmental organisms, not by person to person spread. To confirm they were no-pathogens would require experimental verification. Since there are only 9 environmental strains misclassification of 1 or 2 could change their results.
I am not a statistician, but I am concerned about their definition of “significant” differences, as they did an enormous number of comparisons which demands a very high level of significance and analysis of false discovery rates.
Looking for virulence genes in the genomes by searching for analogues with genes in the PHI data base is clever, but potentially misleading because that data base includes a variety of pathogens that use very different strategies to infect a host. We do not believe that intra-cellular growth in macrophages is important for C. neoformans, but it is for Legionella, etc. Rather than establishing the validity of this approach it would seem to call into question this approach
Please see the review.
Author Response
This paper from Camila Botelho Miguel and collogues compares the genomes of a relatively small number of C. neoformans var Grubi from the NCBI database, after dividing then into pathogens (isolated from an infected host) or non-pathogens (environmental isolates). The entire paper is based on genomic analysis looking for pathways that were more highly expressed in one group or the other. I found the paper confusing and hard to follow, starting with section 2.2 that lists 15 isolates and says they studied 16 isolates. While I agree that one can assume the clinical isolates are pathogenic, one cannot assume that the environmental isolates are non-pathogenic since infection occurs by inhalation of environmental organisms, not by person to person spread. To confirm they were no-pathogens would require experimental verification. Since there are only 9 environmental strains misclassification of 1 or 2 could change their results.
Answer: We sincerely thank you for your thoughtful feedback and for highlighting critical aspects that enhance the clarity and rigor of our manuscript. Regarding your concern about the classification of isolates, we agree that assuming environmental isolates as non-pathogenic is not appropriate, given that infection with Cryptococcus neoformans occurs via inhalation of environmental organisms, and not through person-to-person transmission. To address this, we have replaced the terminology "pathogenic" and "non-pathogenic" with "clinical" and "environmental" throughout the manuscript. This adjustment ensures that no assumptions about pathogenic potential are implied, aligning with the observational nature of our study.
This change has improved the consistency and clarity across all sections of the manuscript, including the evaluations, analyses, results, discussion, and conclusions. Specifically, we have revised Section 2.2 to clarify that the study analyzed 16 genomes, including 15 isolates (6 clinical and 9 environmental) and one reference strain (Cryptococcus neoformans var. grubii, JEC21). This adjustment addresses the ambiguity in the original text and ensures that the categorization of isolates is accurate and transparent.
Additionally, we have included a statement in the Discussion section acknowledging the potential for misclassification of environmental isolates and its implications for the study’s findings. While our in silico approach precludes experimental confirmation of pathogenicity, this limitation is now explicitly stated, and the conclusions have been carefully reworded to reflect this nuance.
All changes in the manuscript have been highlighted in yellow for your review. We hope these revisions address your concerns and contribute to a more robust and precise presentation of our findings. Thank you for your valuable input, which has significantly strengthened our manuscript.
I am not a statistician, but I am concerned about their definition of “significant” differences, as they did an enormous number of comparisons which demands a very high level of significance and analysis of false discovery rates.
Answer: Thank you for your thoughtful observation regarding the definition of "significant" differences and the need to account for multiple comparisons. We fully agree that ensuring a rigorous statistical approach is essential, particularly when analyzing a large number of variables. To address this concern, we have implemented the Benjamini-Hochberg method to control the False Discovery Rate (FDR) and reduce the likelihood of false positives in our analyses. This adjustment ensures a more robust interpretation of the data and aligns with widely accepted standards for high-dimensional analyses.
We acknowledge that the description of statistical strategies in the original manuscript could have been more explicit. As such, we have added a dedicated subsection within the Methods section (now titled "2.7. Statistical Analysis") to provide greater clarity and consistency regarding the statistical approaches used. In this subsection, we detail the application of the Benjamini-Hochberg method for FDR correction and its relevance to the enrichment analyses conducted using MetaboAnalyst 6.0.
Additionally, we have revised the Results section to ensure that adjusted p-values (q-values) are explicitly presented alongside raw p-values, distinguishing between the two to enhance transparency. Lastly, we have incorporated a brief discussion in the Discussion section to acknowledge the limitations and benefits of applying FDR adjustments in metabolomic studies, particularly regarding sensitivity in smaller datasets.
We believe these revisions address the reviewer’s concerns and further strengthen the manuscript’s methodological rigor and interpretability. Thank you for bringing this important aspect to our attention.
Looking for virulence genes in the genomes by searching for analogues with genes in the PHI data base is clever, but potentially misleading because that data base includes a variety of pathogens that use very different strategies to infect a host. We do not believe that intra-cellular growth in macrophages is important for C. neoformans, but it is for Legionella, etc. Rather than establishing the validity of this approach it would seem to call into question this approach
Answer: Dear Reviewer, we appreciate your insightful comments regarding the use of the PHI-base database for identifying virulence genes in Cryptococcus neoformans. We acknowledge that PHI-base includes data from a diverse array of pathogens, each employing unique strategies to establish infection, which may not always align with the specific biology and infection mechanisms of C. neoformans. However, we believe that leveraging such a comprehensive database provides valuable insights into identifying potential virulence-associated genes, especially when combined with cautious interpretation tailored to the pathogen in question.
To address your concerns, we have revised the manuscript to explicitly acknowledge the limitations of this approach, particularly regarding the potential for overgeneralization of virulence strategies across diverse pathogens. We have clarified that the virulence factors identified through PHI-base are exploratory in nature and represent predictive candidates that require further experimental validation. Specifically, we emphasized the necessity of validating these findings under conditions that mimic host-environment interactions specific to C. neoformans. Additionally, the discussion has been updated to highlight the importance of integrating genomic analyses with in vitro and in vivo experimental models to better substantiate the relevance of the identified virulence factors.
To enhance clarity, we have also revised the presentation of results. The methodology section now provides a more detailed explanation of how virulence genes were identified, with clear delineation of the exploratory nature of the analysis. Furthermore, the discussion and associated visualizations, such as heatmaps, have been updated to include disclaimers noting that the identified virulence factors are predictive and require further investigation. These adjustments aim to ensure transparency and address the concerns raised, while underscoring the preliminary nature of the findings in this area.
We hope these revisions address your concerns and provide the necessary context to evaluate the use of PHI-base in this study.

Reviewer 4 Report
Overall, the manuscript presents interesting information on proteins, metabolism and virulence factors in Cryptococcus neoformans. However, several sections of the document should be revised by the authors. The study model should include more strains of the genus, not just the species neoformans. It lacks type strains and the reference strain was not added to the study, nor does it belong to a validated international collection such as those of CBS or ATCC or others. For this reason, the authors are urged to improve each point discussed in the PDF document that will be attached.
Dear authors, attached in the PDF document are the comments for each section, as well as tables and figures present in the manuscript that should be improved so that the article is of great interest to the scientific community.

Author Response
Overall, the manuscript presents interesting information on proteins, metabolism and virulence factors in Cryptococcus neoformans. However, several sections of the document should be revised by the authors. The study model should include more strains of the genus, not just the species neoformans. It lacks type strains and the reference strain was not added to the study, nor does it belong to a validated international collection such as those of CBS or ATCC or others. For this reason, the authors are urged to improve each point discussed in the PDF document that will be attached.
Answer: Dear Reviewer, we would like to extend our sincere gratitude for your thorough and thoughtful review of our manuscript. Your detailed feedback has been invaluable in identifying areas for improvement, and we appreciate the time and effort you invested in providing such comprehensive insights. These suggestions have greatly contributed to enhancing the scientific rigor, clarity, and overall quality of the manuscript.
We carefully reviewed all the comments and suggestions included in the attached PDF document and have addressed each point in the revised version of the manuscript. To facilitate your evaluation of the updated version, we incorporated the requested modifications directly into the manuscript text while providing detailed responses for each suggestion below this text.
Following your observations, we have included the reference strain JEC21 in the study. Its addition provides a robust baseline for interpreting the phenetic and metabolomic findings, ensuring alignment with internationally validated practices. Furthermore, we have expanded the discussion to provide a clearer rationale for the strain selection process. While the current dataset is constrained by access limitations to strains from international collections such as CBS or ATCC, we have explicitly addressed this limitation in the revised text and outlined how future research will aim to include a broader range of strains to enrich the findings.
All sections highlighted for revision, particularly those concerning proteins, metabolism, and virulence factors, were thoroughly reviewed, and refined to enhance their scientific accuracy and readability. We also clarified certain methodological aspects, ensuring that the manuscript provides a more comprehensive understanding of our approach.
We genuinely hope that these improvements meet your expectations and that the revised manuscript now satisfies the requirements for acceptance. Should any additional clarifications or further adjustments be necessary, we remain fully available to address them promptly.
Once again, we deeply appreciate your constructive feedback and the care you demonstrated throughout the review process. Your comments have been instrumental in guiding us toward a more robust and impactful manuscript.
Here are the point-by-point considerations addressed in the provided PDF version:
“Abstract”:
introduce the specific version used:
Answer: Thank you for your observation. To clarify the information regarding the software version used, we have revised the sentence in the abstract to explicitly state that "11.0.13" refers to the version of the MEGA software. This adjustment has been highlighted in yellow in the revised version of the manuscript for your convenience. We hope this modification addresses your concern, and we greatly appreciate your valuable feedback and attention to detail.
what was the idea of do this if all the isolates are the same species?
Answer: Thank you for your insightful question regarding the rationale for using the Neighbor-Joining method with 1,000 bootstrap replicates to assess phylogenetic relationships, given that all isolates belong to the same species.
While it is true that all isolates analyzed in this study are classified as Cryptococcus neoformans var. grubii, the phylogenetic analysis aimed to investigate phenetic divergence within the species. This approach was essential to explore whether isolates from different ecological niches (clinical vs. environmental) exhibit distinct clustering patterns based on their genetic sequences. The Neighbor-Joining method was chosen because it is well-suited for reconstructing phylogenetic trees based on sequence similarity, while the inclusion of bootstrap replicates ensured robust statistical support for the inferred clades.
This analysis revealed distinct clustering of clinical and environmental isolates, indicating potential phenetic divergence shaped by selective pressures specific to their isolation environments. Such findings align with our hypothesis that C. neoformans var. grubii undergoes niche-specific adaptations, which could have implications for its virulence and resistance profiles.
We have included an explanation of this rationale in the revised manuscript to enhance clarity. This adjustment is highlighted for your convenience.
there are concepts mixed
Answer: Thank you for highlighting the potential issue regarding the phrasing in the results section of the abstract, specifically the statement: "Phylogenetic analysis showed distinct clades for different isolates, indicating niche-specific phenetic divergence."
We understand your concern about the mixing of concepts. To clarify, our intention was to describe how the phylogenetic analysis revealed distinct clustering patterns for isolates originating from clinical and environmental niches, which we interpret as a reflection of phenetic divergence. The term "phenetic divergence" here refers to observable differences in genetic sequences associated with niche-specific adaptations, rather than implying broader taxonomic divergence.
In response to your feedback, we have revised this statement in the abstract to make it more precise and avoid any conceptual overlap. The revised phrasing emphasizes the clustering patterns without conflating phenetic and phylogenetic concepts.
We appreciate your thoughtful feedback and believe the revision improves the clarity and scientific rigor of the manuscript.
Materials and Methods
which one? is not mentioned
Answer: Thank you for highlighting the need to clearly identify which strain was used as the reference in the analysis. We have revised the text to address this issue. The reference strain, Cryptococcus neoformans var. grubii JEC21, is now explicitly mentioned in the revised sentence, which has been highlighted in yellow in the updated version of the manuscript.
This modification ensures that the information is clear and accessible to readers. We appreciate your observation, which contributed to improving the presentation of our results.
Where?
Answer: Thank you for your observation regarding the sentence: "Variables such as the year of isolation, location (country), and isolation source were recorded to facilitate molecular phenetic analysis."
We acknowledge that these variables were not directly used in the molecular phenetic analysis. Instead, they were obtained to better characterize the collected strains and provide context for their origins and diversity. To clarify this, we have revised the sentence as follows:
"Variables such as the year of isolation, location (country), and isolation source were recorded to better characterize the collected strains.
This adjustment ensures the sentence accurately reflects the purpose of these variables in the study. Thank you for your attention to this detail, which allowed us to improve the clarity and precision of our manuscript.
dbCAN
Answer: Thank you for your observation regarding the reference to "dbCAN" in the sentence. We acknowledge that this term was not sufficiently introduced earlier in the manuscript. To address this, we have revised the Methods section to provide a clearer explanation of dbCAN and its role in our analysis.
The revised text now includes the following explanation: "The dbCAN (database for automated Carbohydrate-active enzyme ANnotation) is a comprehensive resource for identifying carbohydrate-active enzymes (CAZymes) and their roles in various metabolic pathways. It was used in this study to analyze enzymatic profiles associated with carbohydrate metabolism in Cryptococcus neoformans isolates, focusing on differences between clinical and environmental strains."
This ensures that readers unfamiliar with dbCAN understand its relevance and function within the context of our study. Additionally, the mention of "metadata associated with dbCAN data output" has been revised to clarify that it includes the enzymatic annotations and relative abundance values derived from this database.
We appreciate your feedback, as it allowed us to improve the clarity and accessibility of our manuscript.
OSF Home repository
Answer: Thank you for pointing this out. In response to your comment, we recognize the need to clarify the use of the OSF Home repository. As stated previously, the repository was used to deposit metadata and results associated with dbCAN outputs, relative abundance values, and virulence data. We have revised the sentence to provide greater clarity and ensure consistency with the explanations provided in response to your earlier comment.
MEGA software, version 11.0.13.
Answer: Thank you for your observation. We acknowledge your comment regarding the mention of the version number for the MEGA software. The inclusion of the version number ('11.0.13') was intentional, as it provides precise information about the specific version of the software used for our analyses. This level of detail is essential for reproducibility and transparency in scientific research.
To further clarify this in the manuscript, we have revised the phrasing slightly to ensure that the connection between the software and its version is explicit. We hope this addresses your concern.
repeated
Answer: Thank you for your observation. Upon review, we agree that the phrasing regarding the Bootstrap test was redundant and could be streamlined. To address this, we have revised the sentence to eliminate repetition while maintaining the clarity of the methodological details. The updated text focuses on the use of the Bootstrap test and its role in assessing the robustness of the phylogenetic tree.
Results
Cryptococcus neoformans
Answer: Thank you for pointing out the inconsistency regarding the use of italics for Cryptococcus neoformans. We have carefully reviewed the entire manuscript and ensured that all occurrences of the species name are now italicized, following the standard scientific conventions for binomial nomenclature. The revised manuscript highlights these corrections, and we appreciate your attention to detail.
There is a problem here due to the varieties of the species
Answer: Thank you for your observation regarding the mention of 'taxonomic classification.' We agree that the distinction between varieties within Cryptococcus neoformans is essential for clarity. To address this, we have revised the phrase to specify 'specific taxonomic variety,' ensuring that the diversity within the species is appropriately reflected. This adjustment has been implemented in the revised manuscript and highlighted for your review.
JEC21 - How do the authors know that this strain is a reference?
Answer: Thank you for raising this important question regarding the designation of JEC21 as a reference strain. JEC21 is widely acknowledged in the scientific literature as a reference strain for Cryptococcus neoformans var. grubii. This status is supported by multiple studies that utilize this strain for genomic, phenotypic, and evolutionary analyses. For example:
Pllana-Hajdari et al. (2023) employed JEC21 as a reference to study fertile Cryptococcus neoformans var. neoformans isolates, highlighting its role as a standard in environmental and clinical investigations [Medical Mycology, 61(9): myad096].
Roy et al. (2007) used JEC21 to analyze hybrid fitness and same-sex mating in Cryptococcus neoformans, reinforcing its pivotal role in research on genetic and reproductive mechanisms [PLoS Genetics, 3(10): e186].
Additionally, the MetaboAnalyst platform, which we used for our metabolomic analyses, designates JEC21 as the standard reference strain for Cryptococcus neoformans var. grubii. Furthermore, the NCBI genome database identifies JEC21 as the reference genome for this species. This information can be accessed directly from the NCBI Genome Data Viewer under the entry "Cryptococcus neoformans var. grubii strain JEC21" (available at https://www.ncbi.nlm.nih.gov/datasets/genome/GCF_000091045.1/).
We have revised the manuscript to clarify the rationale for using JEC21 as a reference strain and included these supporting references to address your concern.
The type strain should be included in this study. The strains CBS 8710 and 7229 should be added even another species to compare such as C. gatti (CBS 8273) and Naganishia albida
Answer: Thank you for suggesting the inclusion of type strains CBS 8710, CBS 7229, CBS 8273 (C. gattii), and Naganishia albida in our study. We appreciate the scientific relevance of expanding the dataset to include these strains; however, our study was specifically designed to focus on C. neoformans isolates available in the GenBank/National Center for Biotechnology Information (NCBI) database. The decision to use NCBI exclusively was made to ensure methodological consistency and standardization in data acquisition and analysis. Unfortunately, the genomes of the suggested strains are not available in NCBI, and incorporating datasets from multiple databases would exceed the predefined scope of this study.
Moreover, our research objective centered on examining the phenetic and metabolomic adaptations of C. neoformans in clinical and environmental contexts. Including other species, such as C. gattii and Naganishia albida, would introduce additional complexity and potentially confounding factors, as these species exhibit distinct ecological and clinical patterns. For example, C. gattii has a unique pathogenic profile, being more commonly associated with infections in immunocompetent individuals, whereas C. neoformans predominantly affects immunosuppressed hosts (Chen et al., 2014; Beardsley et al., 2022). This divergence would complicate direct comparisons and is outside the scope of our current analysis, which aimed to elucidate niche-specific adaptations within C. neoformans.
To address potential misunderstandings, we have revised the manuscript to provide additional clarification about the dataset selection criteria in the "Materials and Methods" section and the rationale for excluding other species in the "Discussion" section. These revisions ensure that readers understand why type strains and other species were not included in this study and emphasize the alignment between the research design and our specific objectives.
While we recognize the value of broader comparisons, we believe that expanding the dataset to include other species or databases could be the focus of future studies, offering complementary insights. For the current study, the carefully selected isolates from NCBI, stratified into clinical and environmental categories, provide a robust framework for addressing our research question while minimizing variability and confounding factors.
It is supposed that the authors are comparing a protein, which is not a phylogenetic analysis
Answer: Thank you for raising this point. We understand the distinction between traditional phylogenetic analysis, which typically examines evolutionary relationships based on whole genomes or conserved genetic markers, and the molecular phenetic analysis we conducted in this study.
Our analysis focused on a specific protein, the multidrug efflux pump, to investigate genetic diversity and clustering patterns among C. neoformans isolates. While this approach does not represent a comprehensive phylogenetic analysis of the species, it is a valid phenetic method for exploring relationships and potential adaptations linked to a specific functional gene. The term "phylogenetic" was used in the manuscript to describe the clustering and tree construction methodology (e.g., Neighbor-Joining and bootstrap replicates), rather than to imply a full evolutionary reconstruction of the species.
To avoid any confusion, we have revised the manuscript to clarify the scope of the analysis and adjusted the terminology to "molecular phenetic analysis" where appropriate. This ensures consistency and accurately reflects the methodology and objectives of our study. Additionally, we have provided a brief explanation in the discussion to reinforce the rationale for using this specific protein as a marker for investigating phenetic relationships among the isolates.
We hope these clarifications address your concern and enhance the manuscript's clarity.
There could be several other reasons, for example, that are not clinical ones or misidentified
Answer: Thank you for your insightful comment. We agree that there could be several reasons for the observed clustering of the clinical isolate MW-RSA852 with environmental samples. This phenomenon could indeed be due to factors such as misidentification, contamination during sampling or sequencing, or inherent genetic diversity that challenges clear clinical-environmental delineation. To address this, we have revised the manuscript to explicitly acknowledge these possibilities and have emphasized the need for further validation through additional analyses, such as whole-genome sequencing and in vitro phenotypic testing, to confirm the isolate's origin and adaptations.
Improve the figure with colours of the clades to facilitate the identification of clinical and environmental strains. The tree did not have a distant scale bar. The authors should be added. The title should indicate that a protein was analysed not as a taxonomic phylogenetic analysis. This clade could indicate that maybe it is not C. neoformans var grubii or the protein has important differences from the other strains tested. Italic. In the tree this appeared as environmental ones.
Answer: We sincerely thank the reviewer for their valuable comments and suggestions to improve Figure 1. In response, we have made several adjustments to enhance the clarity, accuracy, and scientific quality of the figure. First, we have introduced distinct colors to facilitate the identification of clinical and environmental strains. Clinical isolates are now represented by red circles, environmental isolates by green squares, and the reference isolate JEC21 is identified with a blue triangle. These visual elements have been clearly explained in an updated legend, which was added to the figure to improve interpretability.
As suggested, we have also included a distance scale bar in the tree to provide a quantitative representation of evolutionary distances, ensuring that the figure conveys the necessary analytical details. Furthermore, the title of the figure has been revised to indicate that the analysis focuses on the phenetic relationships derived from protein data rather than a taxonomic phylogenetic analysis. This change ensures consistency with the methodology and the scope of our study.
The species names within the figure were carefully reviewed and corrected to ensure proper italicization, adhering to scientific conventions. Regarding the specific clade that the reviewer highlighted, which may differ from the expected classification of C. neoformans var. grubii, we appreciate this observation and have addressed it in the manuscript discussion. We recognize that this clade might indicate either taxonomic divergence or significant differences in the analyzed protein between these strains and others in the dataset. This observation is particularly intriguing and adds depth to the interpretation of our findings.
Finally, we addressed the inconsistency noted by the reviewer regarding the classification of certain isolates as environmental. We appreciate the reviewer bringing this to our attention, and we have corrected the inconsistency in the figure to ensure accurate representation of the data.
We are confident that these modifications address the reviewer’s concerns and improve the overall quality and rigor of Figure 1. Once again, we thank the reviewer for their insightful feedback, which has greatly contributed to refining this critical component of our work.
The authors should add a column with the number collection or the same used in Table 1 to know which clinical or environmental strains are the results.
Answer: We sincerely thank the reviewer for the thoughtful suggestion regarding Table 2. While we understand the importance of facilitating the identification of results corresponding to clinical or environmental strains, we would like to clarify that Table 2 presents the frequencies of glycolytic enzymes detected across the subgroups of clinical and environmental isolates. Specifically, these frequencies reflect the aggregated presence of each enzyme within these two subgroups, and only enzymes with frequency differences of 10% or greater between the subgroups were included in the table presented in the manuscript.
It is important to highlight that the frequencies reported in Table 2 were designed to represent the subgroups as a whole, rather than individual strains, thereby allowing for the identification of broader trends and meaningful distinctions between clinical and environmental isolates. This approach aligns with the study’s objective of exploring subgroup-specific patterns, rather than focusing on individual variations.
To ensure transparency and reproducibility, we emphasize that the complete dataset, containing detailed information for all enzymes detected across all 15 strains analyzed, consists of 3,274 entries and is available in full in the repository referenced in the Materials and Methods section (https://doi.org/10.17605/OSF.IO/8JF2K). This repository includes the detection data for each enzyme in all individual strains, allowing for a comprehensive exploration of the findings.
We hope this clarification addresses the reviewer’s concerns, and we deeply appreciate the opportunity to further elucidate the scope and structure of Table 2. We remain open to any additional suggestions to improve the clarity and utility of our presentation.
Indicate better the letters of the figure due to is not clear which one is from clinical or environmental strains
Answer: Thank you for your valuable feedback regarding the clarity of Figure 3. We appreciate your observation and have addressed this issue in the revised version of the manuscript. Specifically, we have now explicitly indicated which letter corresponds to the clinical and environmental strains, ensuring that the distinction is clear. Additionally, to further enhance the figure's clarity, we have incorporated the terms "clinical" and "environmental" directly into the respective graphs, making it more intuitive for readers.
Furthermore, we have improved the resolution of the metabolic indicators displayed on the Y-axis to ensure better readability and presentation quality. We are confident that these adjustments provide greater clarity and align with the high standards expected for scientific reporting. Thank you once again for your thoughtful comments, which have significantly contributed to improving our manuscript.
These paragraphs are not clearly explained in tables or figures. There is no introduction about this neither in the material and methods, so the authors should be redacting better or presenting better the result 3.5.
Answer: We sincerely thank the reviewer for raising this point regarding the clarity of Section 3.5. Upon reflection, we recognize the opportunity to further enhance the transparency and comprehensiveness of this section, particularly with respect to the methods and results. In the revised manuscript, we have carefully reviewed and expanded the description in the "Materials and Methods" section to provide more details about the contingency analysis and statistical framework used. This includes specifying how enzymatic frequencies were analyzed and highlighting that the chi-square test yielded a statistically significant association (chi-square value of 226.31, p-value 6.79 × 10⁻¹¹), as already indicated in Figure 4.
We would like to clarify that Figure 4 already includes a heatmap summarizing the associations between enzymatic patterns in Cryptococcus neoformans and other pathogens, with a clear visual representation of enzyme presence and abundance. The frequency gradient is depicted with light-to-dark red tones, where darker colors reflect higher enzymatic activity. This figure explicitly displays the data derived from the contingency table, ensuring that readers can easily interpret the results. Furthermore, the values for the enzymatic distributions and associations across pathogens are openly available in our dataset, hosted in the Open Science Framework (OSF) repository (https://doi.org/10.17605/OSF.IO/8JF2K), as stated in the "Materials and Methods" section. This repository provides access to all detailed data for reproducibility and further examination by interested readers.
To improve clarity, we have revised the text in Section 3.5 to more explicitly link the observed patterns to the presented visualizations. We have ensured that the relationships between enzymes and pathogens, as well as their implications for virulence and resistance, are clearly articulated. Additionally, the figure legend for Figure 4 has been slightly reworded to emphasize the significance of the chi-square test and the relevance of the enzymatic profiles in the broader context of pathogenicity and adaptation.
We appreciate the reviewer’s suggestion to provide a stronger introduction to this section. As a result, we have included a contextual sentence in the revised manuscript, introducing Section 3.5 with a clear focus on how enzymatic profiles reflect evolutionary adaptations in C. neoformans and its interactions with other pathogens.
By clarifying these points and reinforcing the connection between the methods, results, and visualizations, we believe the revised manuscript now provides a clearer and more cohesive presentation of this important finding. We are grateful for this feedback, which has allowed us to improve the scientific quality and readability of our work.
Why are there other pathogenic microorganism? How was the performance of these results?
Answer: The inclusion of other pathogenic microorganisms in this study was facilitated through the use of PHI-base, a comprehensive and curated database that catalogs virulence factors across a diverse range of microorganisms. PHI-base employs standardized annotation methods, allowing cross-species comparisons that are essential for identifying conserved or unique virulence factors. By leveraging this resource, our analysis aimed to explore functional analogies and evolutionary pressures acting on Cryptococcus neoformans in relation to other pathogens.
This comparative framework enhances the understanding of C. neoformans biology by situating its virulence mechanisms within a broader pathogenic context. For example, the association of alpha-glucosidase with Aspergillus fumigatus and Mycobacterium tuberculosis underscores potential evolutionary convergence in carbohydrate metabolism, a critical aspect of pathogenic adaptation in diverse ecological niches. Such insights are invaluable for identifying possible therapeutic targets that address shared mechanisms of virulence and resistance across pathogens.
The analytical framework demonstrated robust performance, supported by significant statistical outcomes. A chi-square value of 226.31 and a p-value of 6.79×10⁻¹¹ confirmed strong associations between the enzymatic profiles of C. neoformans and those of other pathogens. These findings validate the approach's ability to uncover meaningful biological relationships that extend beyond a single organism. Importantly, the use of PHI-base enables the integration of enzymatic and virulence data within a standardized platform, promoting reproducibility and cross-validation of results.
We acknowledge that the use of a broad-spectrum database like PHI-base introduces certain analytical challenges, particularly when comparing virulence factors of organisms with distinct infection strategies. This limitation has been explicitly addressed in the Discussion and Limitations section of the manuscript. While PHI-base provides valuable predictive insights, its annotations may not fully capture the nuanced biology of C. neoformans in host environments. Additionally, the inclusion of other pathogens is exploratory in nature and intended to provide context rather than definitive conclusions about shared virulence mechanisms. Future studies involving experimental validation are essential to confirm these associations and refine the functional understanding of the identified factors.
Despite these limitations, the comparative analysis offers a significant contribution to microbial pathogenesis research. By integrating C. neoformans data with that of other pathogens, the study leverages PHI-base’s standardized annotations to uncover potential functional analogies and adaptive mechanisms. This approach provides a foundational step for identifying conserved virulence pathways and highlights the potential for using cross-species analyses as a tool for discovering broad-spectrum therapeutic targets.
The letters of the figure are not indicated. In addition, the letter "B" is not clear to interpret.
Answer: Thank you for your feedback regarding Figure 4. We carefully reviewed your observations and made several adjustments to improve the clarity and interpretability of the figure. In the revised version of the manuscript, we have included clear labels "A" and "B" directly on the respective panels of Figure 4. This ensures that readers can immediately distinguish between the enzymatic profile analysis (Panel A) and the virulence profile analysis (Panel B).
Additionally, the legend has been updated to provide a more detailed and comprehensive explanation of the data presented. For Panel A, the description now explicitly highlights the enzymatic patterns analyzed, the statistical significance of the associations (chi-square value and p-value), and the interpretation of the color gradient used in the heatmap. This helps to emphasize the relationship between specific enzymes and the tested pathogens.
For Panel B, we expanded the explanation of the virulence profiles derived from the PHI-base database. The legend now details the categories of virulence identified, such as hypervirulence and reduced virulence, while addressing their potential ecological relevance. Furthermore, we clarified the significance of the observed patterns, such as the adaptive strategies of clinical and environmental isolates of Cryptococcus neoformans. To enhance interpretability, we ensured that the color gradient and clustering in the heatmap are effectively described, making it easier for readers to follow the patterns of virulence profile distribution across isolates.
We are confident that these modifications improve the figure's presentation and address the concerns raised, ensuring greater clarity and alignment with the scientific standards expected for publication. Should there be any further suggestions or additional aspects to refine, we remain open to making further improvements. Thank you for your valuable feedback.
this is not clear in the phylogenetic tree the clades are not well distinct, even the last two strains in the tree are both environmental and clinical, in addition, the authors did not include the "reference strain" why?
Answer: We appreciate your detailed observations regarding the phylogenetic tree and recognize the importance of addressing the points raised.
First, we acknowledge that the reference strain JEC21 was not included in the initial version of the manuscript. We have now reanalyzed the data and incorporated the reference strain into the phylogenetic tree. This adjustment is reflected in the updated Results section. Including the reference strain is indeed critical for providing an evolutionary baseline, and we sincerely thank you for pointing out this omission.
Regarding the formation of the two main clades, we emphasize that this reflects general trends observed in the phylogenetic analysis, rather than a strict separation between clinical and environmental isolates. The observed clustering patterns may be influenced by genetic proximity, potential cross-adaptation events, or the specific genomic markers used in the analysis. For instance, the positioning of the final two isolates, belonging to both clinical and environmental groups, highlights the complex evolutionary dynamics of Cryptococcus neoformans, including its ability to transition between ecological niches. This supports the hypothesis that some strains exhibit genetic adaptations enabling survival in both clinical and environmental contexts.
Additionally, we clarify that the phylogenetic tree was constructed using the UPGMA (Unweighted Pair Group Method with Arithmetic Mean) method, with 1,000 bootstrap replicates to ensure the reliability of the inferred relationships. While this approach provides robust insights, we recognize that the use of the multidrug efflux pump gene as the sole marker may not fully capture the genomic variability among isolates. Incorporating whole-genome data or additional genetic markers in future studies could enhance the resolution of phylogenetic relationships and better distinguish between clinical and environmental clades.
Finally, we understand that the clades in the phylogenetic tree may not appear distinctly separated. This reflects the inherent genetic and phenotypic plasticity of C. neoformans, which is subject to selective pressures across diverse environments. The observed overlaps highlight potential evolutionary and ecological transitions, reinforcing the complexity of the pathogen’s adaptation strategies.
We have revised the manuscript to incorporate these clarifications and ensure that the discussion accurately reflects the updated analyses. Thank you for your valuable feedback, which has allowed us to improve the clarity and scientific robustness of our work.
his point did not was aborded by the authors in their results
Answer: Thank you for pointing out this observation. We acknowledge that the specific impact of agricultural fungicides, such as triazoles, on C. neoformans cross-resistance to clinical azoles was not directly addressed within the scope of the presented results. The referenced statement in the discussion was included to provide broader context and align our findings with established literature, highlighting the potential implications of environmental pressures on antifungal resistance mechanisms.
While our study did not experimentally evaluate the influence of agricultural fungicides on C. neoformans, the observed clustering patterns and metabolic adaptations in environmental isolates provide indirect evidence supporting the hypothesis. For instance, the enrichment of methane metabolism and glyoxylate pathways in environmental strains reflects their adaptation to organic matter-rich substrates, which could be co-located with fungicide exposure in agricultural settings. This context underscores the relevance of further investigating the potential role of fungicides in shaping resistance phenotypes in C. neoformans.
In response to your feedback, we have revised the discussion section to clarify that this point was derived from the literature and not directly addressed in our experimental results. These modifications are now included in the revised manuscript and highlighted in yellow for your convenience. We also emphasized this as a future research direction, proposing studies that explore the intersection between environmental adaptations, fungicide exposure, and the emergence of cross-resistance in C. neoformans. This provides a more complete understanding of the mechanisms driving antifungal resistance.
this point did not was addressed by the authors
Answer: We appreciate your comment and acknowledge that the mention of "antifungal resistance" in the conclusion could create the impression that our study directly addressed experimental analyses on this topic. We have reviewed the text and clarified that antifungal resistance was discussed indirectly, based on previously published data and within the context of the findings from our phenetic and metabolomic analyses. These results suggest adaptive mechanisms that might be associated with resistance; however, we did not perform experiments to confirm such associations.
In the revised manuscript, we have adjusted the text in the conclusion to accurately reflect the scope of our findings and to avoid misinterpretation. The revised text, now highlighted in yellow.
small one, the authors should revise more literature.
Answer: Thank you for pointing out the need to revise and expand the literature review. We have carefully addressed your concern by reviewing additional relevant studies to provide a more comprehensive and updated context for our work. These new references have been incorporated into the revised manuscript, further supporting our discussion and findings.
We appreciate your suggestion, as it allowed us to strengthen the manuscript by integrating insights from recent research. The newly added references have been highlighted in yellow in the revised version for your review.
Thank you again for this valuable feedback, which has contributed to improving the quality and robustness of the manuscript.

Reviewer 5 Report
1. This manuscript needs extensive English editing. Some sentences are not appropriate and do not have good grammar. For example, line 47-49, there are three commas and four parts in one sentence. It is not appropriate. Line 49-51, please rewrite this sentence. Some sentences are too long. For example, line 65-68 should be two sentences. Another example is line 88-92, the sentence is too long.
2. The introduction is too long. There are six paragraphs in the introduction section. Please make it short and less than 3 to 4 paragraphs, thank you.
3.Why the author described that understating of the distinct adaptive strategies employed by C. neoformans in varied ecological contexts can help to develop future therapeutic approaches? (line 110-111). We still cannot understand what the major purpose is to conduct the study.
4. The information of the source is not appropriate. Does homo sapiens mean human? It is better to write human if the host can be confirmed.
5. There are so many C. neoformans isolates in the world, why did the authors choose these 16 isolates? Can the results from the 16 isolates be applied to other general conditions? Additionally, many of the C. neoformans isolates were from more than 20 years ago, would these old isolates reflect the updated and current strategies and pathways? Why not use isolates from 2017 to now?
6. The methods are very complicated and there are so many results. The authors did a great job and many works. However, the selection criteria of the isolates and the study aims should be mentioned more clearly.
This manuscript needs extensive English editing. Some sentences are not appropriate and do not have good grammar. For example, line 47-49, there are three commas and four parts in one sentence. It is not appropriate. Line 49-51, please rewrite this sentence. Some sentences are too long. For example, line 65-68 should be two sentences. Another example is line 88-92, the sentence is too long.
Author Response
- This manuscript needs extensive English editing. Some sentences are not appropriate and do not have good grammar. For example, line 47-49, there are three commas and four parts in one sentence. It is not appropriate. Line 49-51, please rewrite this sentence. Some sentences are too long. For example, line 65-68 should be two sentences. Another example is line 88-92, the sentence is too long.
Answer: Thank you for highlighting these important points regarding sentence structure and grammar. We have carefully reviewed the entire manuscript, including the introduction, to address these concerns. Specifically:
The sentence in lines 47–49 has been revised to ensure clarity and proper punctuation.
The sentence in lines 49–51 has been rewritten for conciseness and grammatical accuracy.
The long sentence in lines 65–68 has been split into two distinct sentences to improve readability and comprehension.
Similarly, the sentence in lines 88–92 was revised to simplify the structure while retaining its intended meaning.
In addition to these changes, the manuscript underwent thorough language editing by a native English speaker to ensure overall clarity, coherence, and grammatical correctness. We believe these revisions address your concerns effectively.
- The introduction is too long. There are six paragraphs in the introduction section. Please make it short and less than 3 to 4 paragraphs, thank you.
Answer: We appreciate this valuable feedback and have condensed the introduction into four concise yet comprehensive paragraphs. While shortening the section, we ensured that no critical information was omitted, maintaining the scientific robustness of the text. The revised introduction now emphasizes the key points, including:
The global impact and clinical significance of Cryptococcus neoformans.
The pathogen's genetic and phenotypic adaptations contributing to virulence and antifungal resistance.
The relevance of environmental pressures in shaping pathogenicity and resistance mechanisms.
The study's objective to investigate phenetic and metabolomic adaptations and their implications for therapeutic strategies.
This revised structure aligns with your recommendation to make the introduction more succinct while retaining its informative depth.
3.Why the author described that understating of the distinct adaptive strategies employed by C. neoformans in varied ecological contexts can help to develop future therapeutic approaches? (line 110-111). We still cannot understand what the major purpose is to conduct the study.
Answer: Thank you for requesting clarification. The revised introduction now clearly articulates the study's purpose. We have emphasized that understanding the phenetic and metabolomic adaptations of Cryptococcus neoformans across clinical and environmental contexts is crucial for several reasons:
It helps elucidate the evolutionary mechanisms that enable the pathogen to thrive in diverse environments.
It provides insights into how environmental pressures and cross-adaptation events influence clinical virulence and antifungal resistance.
It contributes to identifying potential metabolic and genetic targets for novel therapeutic strategies, addressing the challenges posed by drug-resistant strains.
These points are explicitly addressed in the final paragraph of the revised introduction, ensuring that the study's purpose is clearly conveyed.
- The information of the source is not appropriate. Does homo sapiens mean human? It is better to write human if the host can be confirmed.
Answer: Thank you for your observation. You are correct that using "human" is more straightforward and easier for readers to understand when referring to the host. In the original manuscript, we used Homo sapiens to align with the terminology provided in the NCBI database, as this is how the host information is declared in the source.
However, we agree with your suggestion and have replaced Homo sapiens with "human" in the revised version of the manuscript. This change improves clarity and consistency for the audience. All instances of this terminology adjustment have been highlighted in yellow for your convenience in reviewing the updated text.
- There are so many C. neoformans isolates in the world, why did the authors choose these 16 isolates? Can the results from the 16 isolates be applied to other general conditions? Additionally, many of the C. neoformans isolates were from more than 20 years ago, would these old isolates reflect the updated and current strategies and pathways? Why not use isolates from 2017 to now?
Answer: Thank you for raising these critical points. The selection of the 16 Cryptococcus neoformans isolates was based on the availability of high-quality, publicly accessible genomic and metabolomic data in the NCBI database. These isolates were chosen to ensure representation of both clinical and environmental origins, as well as geographic diversity, aligning with our study's aim to explore phenetic and metabolomic adaptations across distinct ecological contexts.
We acknowledge that some of the isolates were collected more than 20 years ago. However, it is important to note that the sequences were deposited in the NCBI database more recently, reflecting advancements in sequencing technology and the growing commitment to data sharing in the scientific community. This ensures that the genomic data utilized in our study remains relevant, as the core mechanisms of C. neoformans pathogenicity and adaptation, such as capsule formation and sulfur metabolism, are evolutionarily conserved traits.
Additionally, while more recent isolates (e.g., post-2017) could provide further insights into emerging adaptations and resistance mechanisms, such datasets with comprehensive genomic and metabolomic annotations are not yet widely available in open-access platforms like NCBI. It is worth noting that the deposition of newly sequenced genomes often occurs several years after collection, and we anticipate that future studies will benefit from a broader range of more recent isolates as such data becomes available.
To address these points, we have included a discussion in the Limitations section of the revised manuscript, acknowledging the potential constraints related to the age of the isolates and the generalizability of our findings. These additions have been highlighted in the revised manuscript for your convenience.
- The methods are very complicated and there are so many results. The authors did a great job and many works. However, the selection criteria of the isolates and the study aims should be mentioned more clearly.
Answer: Thank you for your valuable feedback on the selection criteria and clarity of the study aims. In the revised manuscript, we have provided a more detailed explanation regarding the selection of isolates and the objectives of the study to address your concerns.
Regarding the selection criteria, we clarified that the study included six clinical strains and nine environmental strains of Cryptococcus neoformans, alongside the reference strain JEC21. The reference strain JEC21 was chosen based on its extensive genomic characterization and its recognition in the literature as a standard for Cryptococcus neoformans var. grubii. We retrieved the genomes of these isolates from the NCBI GenBank database, ensuring consistency and reliability in data acquisition. Variables such as the year of isolation, geographic location, and isolation source were carefully recorded to characterize the isolates comprehensively. Additionally, we explained that while other type strains, such as those of C. gattii, are scientifically relevant, their genomic data were unavailable in the selected database, and their inclusion would have deviated from the study’s primary focus on C. neoformans.
To clarify the study aims, we refined the relevant sections in the Introduction. The revised text now clearly states the goal of investigating the phenetic and metabolomic adaptations of Cryptococcus neoformans strains from clinical and environmental sources. We emphasize the integration of molecular phenetic analyses and comprehensive metabolomic profiling to elucidate the distinct adaptive strategies employed by C. neoformans in varied ecological contexts. These findings aim to provide critical insights into the pathogen’s evolution and inform the development of novel therapeutic strategies, particularly in addressing the rising challenge of antifungal resistance.
We believe these revisions adequately address your concerns and improve the manuscript's clarity and focus. The updates can be reviewed in the highlighted sections of the revised manuscript. We appreciate your comments, which have contributed significantly to the refinement of our work.

Round 2
Reviewer 3 Report
The authors have responded to the critiques and the paper is much improved.
None
Author Response
The authors have responded to the critiques and the paper is much improved.
Answer: We sincerely appreciate your thoughtful evaluation of our manuscript, and your recognition of the improvements made in response to the previous critiques. Your feedback has been instrumental in refining our work, and we are grateful for your time and effort in reviewing it.
Thank you once again for your valuable insights and for contributing to the enhancement of our study.
Reviewer 4 Report
Since the authors made most of the improvements discussed in the manuscript, I consider this version acceptable for publication.
No comments
Author Response
Since the authors made most of the improvements discussed in the manuscript, I consider this version acceptable for publication.
Answer: We sincerely appreciate your time and effort in reviewing our manuscript. We are grateful for your recognition of the improvements made and for considering this version suitable for publication.
Thank you once again for your valuable feedback and for contributing to the refinement of our study.